# Passive accumulation of alkaloids in inconspicuously colored frogs refines the evolutionary paradigm of acquired chemical defenses

Rebecca D Tarvin[1]*, Jeffrey L Coleman[2,3], David A Donoso[4,5], Mileidy Betancourth-Cundar[6,7], Karem López-Hervas[8], Kimberly S Gleason[9], J Ryan Sanders[9], Jacqueline M Smith[9], Santiago R Ron[10], Juan C Santos[11], Brian E Sedio[2,3], David C Cannatella[2]*, Richard W Fitch[9]*

[1]Museum of Vertebrate Zoology and Department of Integrative Biology, University of California, Berkeley, Berkeley, United States; [2]Department of Integrative Biology and Biodiversity Collections, University of Texas at Austin, Austin, United States; [3]Smithsonian Tropical Research Institute, Ancón, Panama; [4]Grupo de Investigación en Ecología Evolutiva en los Trópicos (EETROP), Universidad de las Américas, Quito, Ecuador; [5]Ecological Networks Lab, Technische Universität Darmstadt, Darmstadt, Germany; [6]Departamento de Ciencias Biológicas, Universidad de los Andes, Bogotá, Colombia; [7]Department of Biology, Stanford University, Palo Alto, United States; [8]Max Planck Institute for Evolutionary Biology, Plön, Germany; [9]Department of Chemistry and Physics, Indiana State University, Terre Haute, United States; [10]Museo de Zoología, Escuela de Biología, Facultad de Ciencias Exactas y Naturales, Pontificia Universidad Católica del Ecuador, Quito, Ecuador; [11]Department of Biological Sciences, St John's University, New York, United States

*For correspondence:
rdtarvin@berkeley.edu (RDT);
catfish@utexas.edu (DCC);
Richard.Fitch@indstate.edu (RWF)

Competing interest: The authors declare that no competing interests exist.

## eLife Assessment

This study is **important**, with the potential to greatly impact future research on the evolution of chemical defense mechanisms in animals. The authors present **compelling** evidence for the presence of low quantities of alkaloids in amphibians previously thought to lack these toxins. They then integrate these findings with existing literature to propose a four-phase scenario for the evolution of chemical defense in alkaloid-containing poison frogs, emphasizing the role of passive accumulation mechanisms.

**Abstract** Understanding the origins of novel, complex phenotypes is a major goal in evolutionary biology. Poison frogs of the family Dendrobatidae have evolved the novel ability to acquire alkaloids from their diet for chemical defense at least three times. However, taxon sampling for alkaloids has been biased towards colorful species, without similar attention paid to inconspicuous ones that are often assumed to be undefended. As a result, our understanding of how chemical defense evolved in this group is incomplete. Here, we provide new data showing that, in contrast to previous studies, species from each undefended poison frog clade have measurable yet low amounts of alkaloids. We confirm that undefended dendrobatids regularly consume mites and ants, which are known sources of alkaloids. Thus, our data suggest that diet is insufficient to explain the defended phenotype. Our data support the existence of a phenotypic intermediate between toxin consumption

and sequestration — passive accumulation — that differs from sequestration in that it involves no derived forms of transport and storage mechanisms yet results in low levels of toxin accumulation. We discuss the concept of passive accumulation and its potential role in the origin of chemical defenses in poison frogs and other toxin-sequestering organisms. In light of ideas from pharmacokinetics, we incorporate new and old data from poison frogs into an evolutionary model that could help explain the origins of acquired chemical defenses in animals and provide insight into the molecular processes that govern the fate of ingested toxins.

## Introduction

### Overview

Complex phenotypes can evolve by leveraging phenotypic plasticity in existing traits with concerted change in developmental modules (*West-Eberhard, 2003*). However, the evolutionary trajectory that animals take to traverse an adaptive landscape from one phenotype to another may be difficult to reconstruct given that they often must cross or avoid adaptive valleys, which include phenotypes that are not always readily observed in populations (e.g. *Martin and Wainwright, 2013*). Nevertheless, phenotype diversity can help us unravel origins of novel traits and reveal the physiological trade-offs associated with their evolutionary trajectory (*Tarvin et al., 2017*).

Acquired chemical defenses, or the ability to sequester and use chemicals from the environment against predators or parasites, is one complex phenotype whose evolutionary history has proved difficult to characterize (*Berenbaum, 1995*; *Santos et al., 2016*). Although human interest in poisonous plants and animals is old — dating back millennia (*Charitos et al., 2022*) — we have only recently begun to elucidate the specific mechanisms involved in acquired chemical defenses (*Beran and Petschenka, 2022*). This persisting gap in knowledge may be partly explained by a historical lack of integration between systems biology and pharmacology (*Rostami-Hodjegan, 2012*). Here, we incorporate ideas from pharmacokinetics with data from poison frogs (Anura: Dendrobatidae) into an evolutionary model that could help explain the origins of acquired chemical defenses in poison frogs and more generally in other animals.

In the following text, we use the terms alkaloid and toxin interchangeably, although the toxicity of each poison frog alkaloid is not always known or very straightforward (*Lawrence et al., 2023*). Similarly, for simplicity we broadly bin species as defended (high alkaloid content) or undefended (low or zero alkaloid content), although little information exists regarding the defensive efficacy of specific alkaloids. In this context, we use the term alkaloid to refer to compounds with nitrogen-containing rings, specifically the subset of lipophilic alkaloids representing classes previously described in anuran integument, for example 'N-methyldecahydroquinolines' or 'lehmizidines' (e.g. *Daly et al., 2009*; *Daly et al., 2005*).

### The history of research leading to the current paradigm: the diet-toxicity hypothesis

In the 1980s, Toft characterized several types of foraging behaviors in neotropical frogs and found that active foraging for ants was common in poisonous frogs (Dendrobatidae and Bufonidae), while sit-and-wait predation on larger prey was common in non-poisonous species (*Toft, 1981*; *Toft, 1980*). Toft hypothesized that chemical defenses protected poisonous species from the greater predation risk incurred by active foraging. At the time, it was thought that poisonous dendrobatids synthesized their own alkaloids (the biosynthetic hypothesis; reviewed by *Saporito et al., 2009*), so differences in diet were not considered mechanistically relevant to differences in levels of chemical defense. However, *Daly et al., 1994a* later demonstrated that chemically defended dendrobatid frogs obtained alkaloids from their diet. This dietary hypothesis led researchers to reevaluate the evolutionary importance of active foraging and hypothesize that specialization on ants promoted the evolution of chemical defense in Dendrobatidae (*Caldwell, 1996*). Later, a more detailed phylogenetic analysis of Dendrobatidae revealed that chemical defense and diet specialization co-evolved independently several times (*Santos et al., 2003*). The new information helped generate the diet-toxicity hypothesis, which posits that shifts from a generalist to a specialist diet are correlated with origins of alkaloid uptake (*Darst et al., 2005*; *Santos and Cannatella, 2011*). Since then, many studies have focused on the

**eLife digest** For most animals, the ability to deter predators is vital for survival. Some organisms, such as poison frogs, use bad tasting or toxic chemicals to ward off predators. In the 1990s, scientists discovered that poison frogs acquire their defensive alkaloid chemicals from the mites, ants and other arthropods they eat.

Many poison frog species use bright or contrasting colors to advertise their defenses to predators; this strategy is known as 'aposematism'. Aposematic frogs have evolved biochemical mechanisms to transport, store and even modify the alkaloid toxins. Although aposematism has evolved independently in three poison frog clades, most of the frogs in this family are dull-colored. These dull-colored frogs are generally assumed to not be able to accumulate alkaloid toxins from their diet. However, very little is known about how animals evolve to be able to use chemicals they eat as toxins to defend themselves.

To learn more about this phenomenon, Tarvin et al. screened different 'undefended' frog lineages for alkaloids to determine whether the frogs lacked them, as previously assumed. The researchers used highly sensitive chromatography and mass spectrometry, techniques that can detect and identify specific compounds in chemical mixtures, even at very low concentrations.

The results showed that nearly every 'undefended' poison frog had alkaloids, but at substantially lower levels than aposematic species. Tarvin et al. propose that these frogs do not have the transport or storage systems that aposematic frogs employ to use the toxic alkaloids they consume. Rather, the dull-colored frogs accumulate alkaloids passively. This 'passive accumulation' appears to be a stepping stone on the path to evolving the ability to accumulate toxins from the diet. Tarvin et al. also found that all of the studied poison frogs ate ants and mites, the main arthropod sources of alkaloids in poison frogs.

The findings of Tarvin et al. suggest that specialized diets are not enough to explain how poison frogs evolved the ability to accumulate toxins. Changes in toxin absorption, distribution, metabolism and excretion are also required for frogs to be able to use alkaloids from their diet as poison. This provides new insights into the evolution of chemical defense in poison frogs and could help researchers to better understand how this type of defense evolved in other animals.

diet of poison frogs in an effort to directly connect diet with chemical defense in specific species (e.g. *McGugan et al., 2016*; *Osorio et al., 2015*; *Sanches et al., 2023*; *Sánchez-Loja et al., 2024*) and to identify sources of poison frog alkaloids (e.g. *Saporito et al., 2007b*; *Saporito et al., 2004*). In general, most of the studies of poison-frog ecology since the 1990s emphasize or assume that diet is a primary determinant of defense.

## A new paradigm: the passive-accumulation hypothesis

Although in the 1990s Daly and his colleagues proposed that an alkaloid uptake system was present in the ancestor of Dendrobatidae and is overexpressed in aposematic species (*Daly, 1998*; *Daly et al., 1994b*; *Saporito et al., 2009*), no details about this purported system were given, and little focus was placed on the physiological processes of alkaloid sequestration in poison frogs for nearly 20 years. *Santos et al., 2016* noted that the study of acquired chemical defenses is 'essentially a study in pharmacokinetics'. Pharmacokinetics (or toxicokinetics, for toxins; *Spurgeon et al., 2020*) is the study of how bioactive compounds are processed by animals. Organismal processes are often binned into four categories together known as ADME, which stands for Absorption, or movement into the bloodstream, Distribution, or movement into and out of body compartments, Metabolism, or biotransformation of the compound, and Excretion, or elimination from the body (*Ruiz-Garcia et al., 2008*). Herein we use similar terms that are more directly relevant to the study of acquired chemical defenses: toxin intake, or the amount of toxin consumed; toxin elimination, or the metabolic detoxification and/ or elimination of toxins from the body (equivalent to Metabolism +Excretion); toxin sequestration, or the transport and storage of toxins to a specific location such as the skin (a modified version of Distribution); and toxin accumulation, or the retention of toxins in an animal, whether or not it is by sequestration processes.

Applying ideas from pharmacokinetics to acquired chemical defenses leads us to propose a four-phase evolutionary model, which we call the passive-accumulation hypothesis: (1) consistent exposure to a toxic compound; (2) prior existence or evolution of some resistance to the toxin; (3) change in the elimination rate of the compound that leads to its prolonged retention, hereafter passive accumulation; and (4) adaptation of molecular pathways to transport and store the compound in a specific location, hereafter sequestration, which results in the chemical defense phenotype. Phases 3 and 4 may both select for increased toxin resistance, initiating a positive feedback loop that could intensify chemical defense and resistance over time. Note that while we focus on the physiological processes underlying toxin resistance and sequestration, other selection pressures including predators may influence these patterns (Other factors that may shape the evolution of acquired chemical defenses).

*Savitzky et al., 2012* defined 'sequestration' as 'the evolved retention within tissues of specific compounds, not normally retained in the ancestors of the taxon in question, which confers a selective advantage through one or more particular functions'. We define passive accumulation as a type of toxin accumulation that is temporary and results from the delay between toxin intake and elimination; an example would be the temporary accumulation then clearance of ibuprofen in blood plasma in humans following ingestion (*Albert and Gernaat, 1984*). We differentiate passive accumulation from sequestration, a term that we argue implies the existence of a derived form of a transport or storage mechanism absent in the ancestor of the taxon, which would permit greater levels of and more long-term toxin accumulation than passive accumulation. In other systems such as insects, mechanisms of sequestration are sometimes described as passive (occurring by diffusion) or active (energy-consuming; *Petschenka and Agrawal, 2016*). Given the general lack of data regarding the mechanisms underlying sequestration in frogs, we refrain from applying these modifiers to the sequestration term.

To develop and refine this hypothesis, we gathered diet and toxin data from a broad selection of aposematic and inconspicuously colored poison-frog species. Approximately 100 of the 345 dendrobatid poison-frog species (*AmphibiaWeb, 2023*) fall into three conspicuously colored and alkaloid-sequestering (aposematic) clades: *Ameerega*, *Epipedobates*, and Dendrobatinae. The other 245 species compose several other primarily inconspicuously colored clades that for the most part have been assumed to lack alkaloid defenses: that is, all Aromobatinae (e.g. *Allobates*, *Rheobates*, *Anomaloglossus*, and *Aromobates*), all Hyloxalinae (*Ectopoglossus*, *Hyloxalus*, *Paruwrobates*), and some Colostethinae (*Colostethus*, *Silverstoneia*, *Leucostethus*; *Figure 1*). According to the phylogenetic placement of defended and undefended species within Dendrobatidae, poison frogs have evolved sequestration of lipophilic alkaloids from consumed arthropods at least three times (*Santos et al., 2014*; *Santos et al., 2003*), making them a suitable group to study complex phenotypic transitions like the evolution of chemical defense.

In total, we surveyed 104 animals representing 32 species of Neotropical frogs including 28 dendrobatid species, 2 bufonids, 1 leptodactylid, and 1 eleutherodactylid (see Methods). Each of the major undefended clades in Dendrobatidae (*Figure 1*, *Table 1*) is represented in our dataset, with a total of 14 undefended dendrobatid species surveyed. Next, we review old and new evidence from poison frogs in the context of the four-phase model (Phases 1 and 2: Consistent exposure to toxins may select for resistance in poison frog and Phases 3 and 4: Evidence for passive accumulation and sequestration in poison frogs). Then we describe major predictions that need further testing to validate and/or revise the proposed model (Predictions arising from the passive-accumulation hypothesis). Finally, we discuss other factors that might influence the evolution of chemical defenses (Other factors that may shape the evolution of acquired chemical defenses), the passive accumulation phenotype in a broader evolutionary context (The passive-accumulation phenotype in a broader evolutionary context), and possible limitations of this study (Limitations). Overall, we propose that further integrating ideas from pharmacokinetics into studies of acquired chemical defenses will lead to new insight in the field, with clear applications to human and ecosystem health. In that vein, we suggest that evolutionary changes in toxin resistance and metabolism are critical physiological shifts that facilitate origins of acquired chemical defenses in animals.

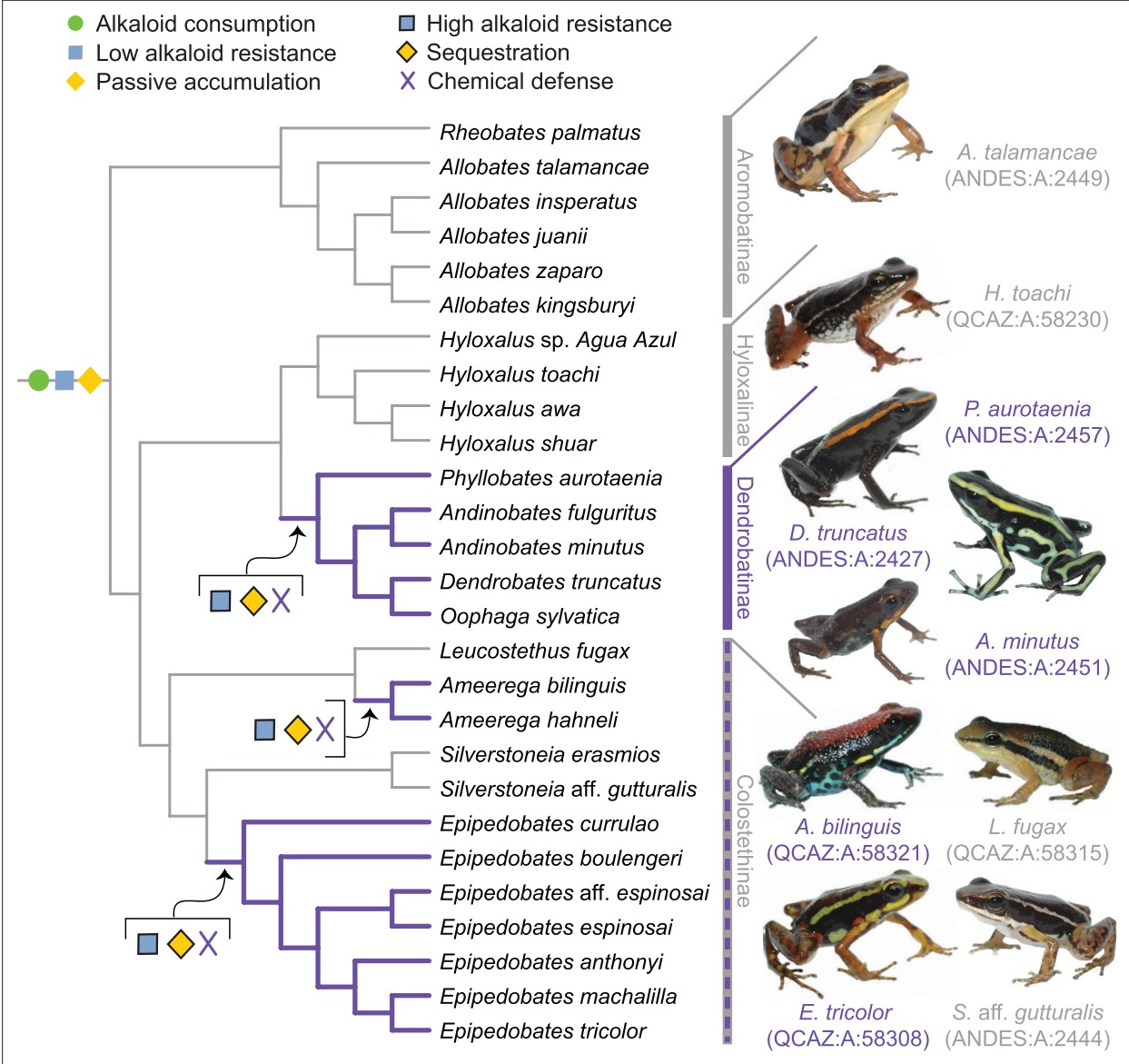

**Figure 1.** A new evolutionary model of toxin sequestration in Dendrobatidae. We propose that alkaloid consumption, some level of alkaloid resistance, and passive accumulation were present in the most recent common ancestor of Dendrobatidae; enhanced resistance and sequestration mechanisms then arose later, resulting in the chemical defense phenotype. Our model places less emphasis on dietary changes compared to prior studies, and more strongly emphasizes novel molecular mechanisms (e.g. binding proteins and target-site insensitivity; *Alvarez-Buylla et al., 2023*; *Tarvin et al., 2017*; *Tarvin et al., 2016*). Purple lines indicate lineages with chemical defense. Gray lines indicate lineages that putatively lack chemical defense. All images of frogs were taken by RDT and are identified by their museum number.

## Results and discussion

### Phases 1 and 2: Consistent exposure to toxins may select for resistance in poison frogs

Several of the lipophilic alkaloids found in dendrobatid frogs have been traced to arthropod sources, specifically mites (*Saporito et al., 2007b*), ants (*Saporito et al., 2004*), and beetles (*Dumbacher et al., 2004*), although the extent to which such arthropod prey vary in alkaloid diversity and quantity remains relatively unstudied. Regardless, broad-scale shifts in diet content towards a higher proportion of ants and mites have been hypothesized to play an important role in the origin of chemical defense in poison frogs (*Darst et al., 2005*; *Santos and Cannatella, 2011*).

**Table 1.** Range and median of alkaloid quantity (estimated by the sum of integrated areas) and alkaloid diversity (number of different compounds) by species from the GC-MS assessment.

The presumed chemical defense phenotype for poison frogs is given according to *Santos and Cannatella, 2011*. Purple rows highlight defended species. *From a UHPLC-HESI-MS/MS dataset for which alkaloids were not quantified. Note that the UHPLC-HESI-MS/MS and GC-MS assays differed in both instrument and analytical pipeline, so 'Alkaloid Number' values from the two assay types should not be compared to each other directly.

| Family | Subfamily | Species | Phenotype | Sample Size (frogs) | Log (Total Integrated Area) Range | Median | Alkaloid Number Range | Median |
|---|---|---|---|---|---|---|---|---|
| Dendrobatidae | Aromobatinae | *Rheobates palmatus* | undefended | 4 | 13.07–14.24 | 14.00 | 1–4 | 1.5 |
| Dendrobatidae | Aromobatinae | *Allobates insperatus* | undefended | 8 | 13.47–15.44 | 14.99 | 1–9 | 5.0 |
| Dendrobatidae | Aromobatinae | *Allobates juanii* | undefended | 1 | 14.10 | 14.10 | 1 | 1.0 |
| Dendrobatidae | Aromobatinae | *Allobates kingsburyi* | undefended | 1 | 13.63 | 13.63 | 2 | 2.0 |
| Dendrobatidae | Aromobatinae | *Allobates talamancae* | undefended | 3 | 14.89–16.27 | 15.09 | 2–4 | 3.0 |
| Dendrobatidae | Aromobatinae | *Allobates zaparo* | undefended | 1 | 16.78 | 16.78 | 8 | 8.0 |
| Dendrobatidae | Colostethinae | *Leucostethus fugax* | undefended | 8 | 12.57–15.33 | 14.00 | 3–8 | 4.5 |
| Dendrobatidae | Colostethinae | *Ameerega bilinguis* | defended | 1 | 21.97 | 21.97 | 133 | 133.0 |
| Dendrobatidae | Colostethinae | *Ameerega hahneli* | defended | 4 | 20.21–22.29 | 21.68 | 85–140 | 128.5 |
| Dendrobatidae | Colostethinae | *Silverstoneia flotator** | undefended | 12 | NA | NA | 0–1 | 0.0 |
| Dendrobatidae | Colostethinae | *Silverstoneia* aff. *gutturalis* | undefended | 9 | 11.80–17.33 | 15.40 | 1–10 | 3.0 |
| Dendrobatidae | Colostethinae | *Silverstoneia erasmios* | undefended | 2 | 14.70–16.11 | 15.41 | 15–15 | 15.0 |
| Dendrobatidae | Colostethinae | *Epipedobates* aff. *espinosai* | defended | 2 | 18.44–20.20 | 19.32 | 83–131 | 107.0 |
| Dendrobatidae | Colostethinae | *Epipedobates anthonyi* | defended | 1 | 20.54 | 20.54 | 127 | 127.0 |
| Dendrobatidae | Colostethinae | *Epipedobates boulengeri* | defended | 2 | 18.87–19.39 | 19.13 | 77–94 | 85.5 |
| Dendrobatidae | Colostethinae | *Epipedobates currulao* | defended | 2 | 19.49–19.68 | 19.59 | 99–105 | 102.5 |
| Dendrobatidae | Colostethinae | *Epipedobates espinosai* | defended | 2 | 18.82–21.33 | 20.08 | 85–146 | 115.5 |
| Dendrobatidae | Colostethinae | *Epipedobates machalilla* | defended | 2 | 12.98–15.67 | 14.32 | 8–38 | 23.0 |
| Dendrobatidae | Colostethinae | *Epipedobates tricolor* | defended | 2 | 18.36–19.07 | 18.72 | 91–114 | 102.5 |
| Dendrobatidae | Hyloxalinae | *Hyloxalus awa* | undefended | 7 | 0.00–16.05 | 13.58 | 0–12 | 3.0 |
| Dendrobatidae | Hyloxalinae | *Hyloxalus shuar* | undefended | 1 | 14.92 | 14.92 | 5 | 5.0 |
| Dendrobatidae | Hyloxalinae | *Hyloxalus* sp. Agua Azul | undefended | 1 | 14.30 | 14.30 | 8 | 8.0 |
| Dendrobatidae | Hyloxalinae | *Hyloxalus toachi* | undefended | 2 | 0.00–0.00 | 0.00 | 0–0 | 0.0 |
| Dendrobatidae | Dendrobatinae | *Phyllobates aurotaenia* | defended | 4 | 17.72–21.08 | 18.88 | 48–118 | 67.5 |
| Dendrobatidae | Dendrobatinae | *Dendrobates truncatus* | defended | 3 | 20.05–23.95 | 20.42 | 111–172 | 115.0 |
| Dendrobatidae | Dendrobatinae | *Oophaga sylvatica* | defended | 5 | 22.86–24.85 | 23.76 | 152–189 | 175.0 |
| Dendrobatidae | Dendrobatinae | *Andinobates fulguritus* | defended | 2 | 20.09–20.51 | 20.30 | 80–85 | 82.5 |
| Dendrobatidae | Dendrobatinae | *Andinobates minutus* | defended | 4 | 16.57–18.77 | 18.07 | 34–80 | 66.0 |
| Bufonidae | | *Amazophrynella siona* | NA | 2 | 14.12–14.40 | 14.26 | 1–1 | 1.0 |
| Bufonidae | | *Atelopus* aff. *spurrelli* | NA | 1 | 11.58 | 11.58 | 4 | 4.0 |
| Eleutherodactylidae | | *Eleutherodactylus cystignathoides** | NA | 3 | NA | NA | 0–0 | 0.0 |
| Leptodactylidae | Leptodactylinae | *Lithodytes lineatus* | NA | 2 | 0.00–0.00 | 0.00 | 0–0 | 0.0 |

We quantified gut contents for 32 species of Neotropical frogs. Both undefended and defended dendrobatid species consume a large proportion of ants and mites (*Figure 2*; *Supplementary file 1*). Although the defended dendrobatid clades tend to consume proportionally more ants and mites, as in other studies, the undefended lineages do consume a high proportion of ants and

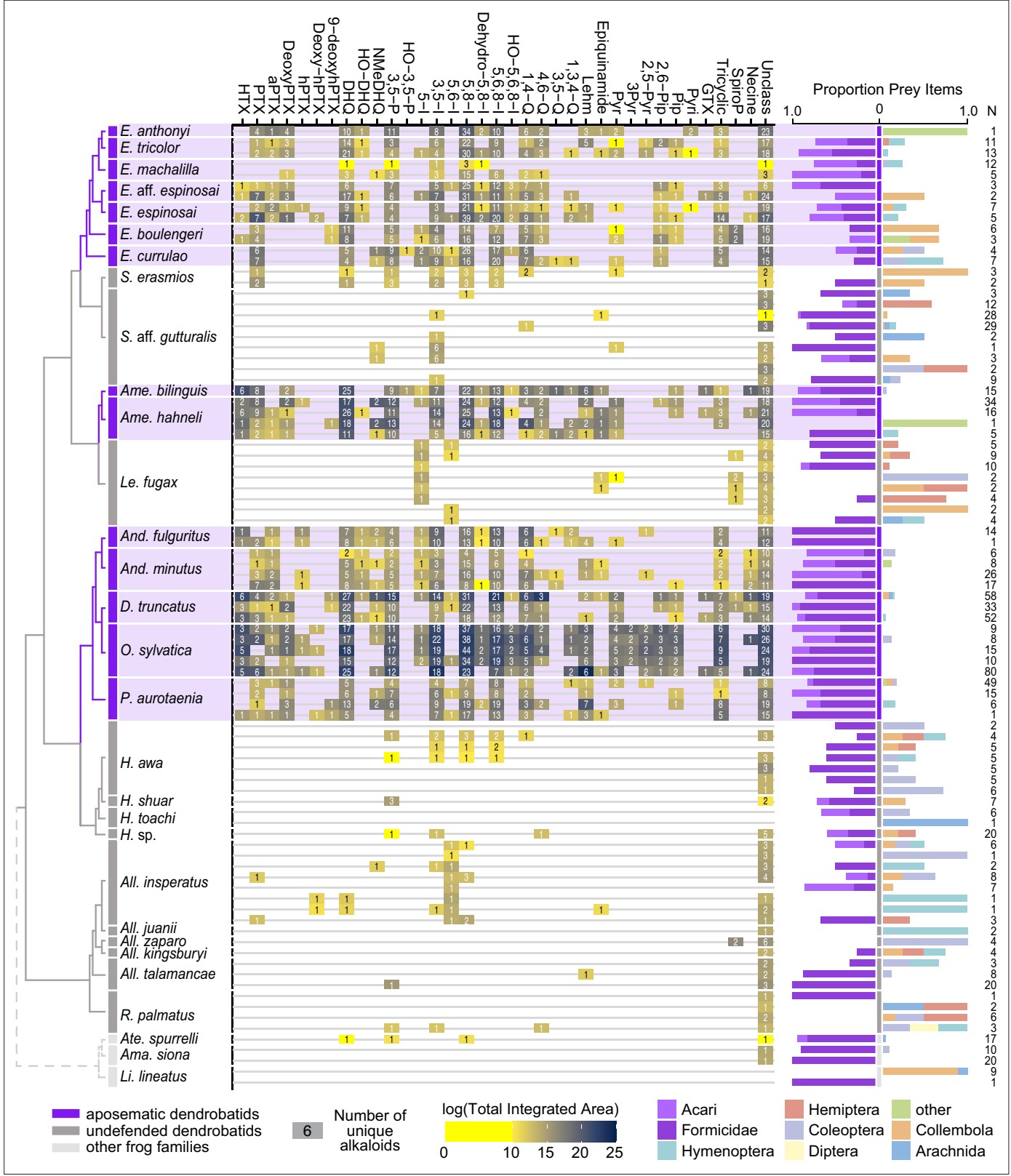

**Figure 2.** From left to right: an ultrametric tree showing phylogenetic relationships inferred previously (*Wan et al., 2023*) among sampled species with the three defended poison frog clades highlighted in purple, the undefended clades in dark gray, and non-dendrobatids in light gray (Bufonidae: *Amazophrynella siona* and *Atelopus* aff. *spurrelli*; Leptodactylidae: *Lithodytes lineatus*). Tile color indicates the log of the total quantity of alkaloids in each class as measured by the sum of integrated areas of alkaloids of that class from GC-MS data per individual. The number in each tile indicates

*Figure 2 continued on next page*

*Figure 2 continued*

the number of alkaloids (including isomers) detected in each individual for each class. On the right are prey items recovered from the stomach of each individual, colored by arthropod group and scaled to 1 (total number of prey identified are shown under N). Note the large proportion of ants (Formicidae, dark purple) and mites (Acari, light purple) in many of the individuals compared to other prey types. See *Supplementary file 1* for raw diet data and *Supplementary file 4* for full alkaloid data. Poison-frog genera names are abbreviated as follows: *All.*, *Allobates*; *Ame.*, *Ameerega*; *And.*, *Andinobates*; *D.*, *Dendrobates*; *E.*, *Epipedobates*; *H.*, *Hyloxalus*; *Le.*, *Leucostethus*; *O.*, *Oophaga*; *P.*, *Phyllobates*; *R.*, *Rheobates*; *S.*, *Silverstoneia*; Alkaloid class abbreviations are based on *Daly et al., 2009*; *Daly et al., 2005* and are as follows: HTX, histrionicotoxins; PTX, pumiliotoxins; PTXB, pumiliotoxin B; aPTX, allopumiliotoxins; DeoxyPTX, deoxypumiliotoxins; hPTX, homopumiliotoxins; deoxy-hPTX, deoxy-homopumiliotoxins; DHQ, decahydroquinolines; NMeDHQ, N-methyldecahydroquinolines; HO-DHQ, hydroxy-decahydroquinolines; 3,5 P, 3,5-disubstituted pyrrolizidines; HO-3,5-P, hydroxy-3,5-disubstituted pyrrolizidines; 5-I, 5-substituted indolizidines; 3,5-I, 3,5-disubstituted indolizidines; 5,6-I, 5,6-disubstituted indolizidines; 5,8-I, 5,8-disubstituted indolizidines; Dehydro-5,8-I, dehydro-5,8-indolizidines; 5,6,8-I, 5,6,8-trisubstituted indolizidines; HO-5,6,8-I, hydroxy-5,6,8-trisubstituted indolizidines; 1,4-Q, 1,4-disubstituted quinolizidines; 4,6-Q, 4,6-disubstituted quinolizidines; 3,5-Q, 3,5-disubstituted quinolizidines; 1,3,4-Q, 1,3,4-trisubstituted quinolizidines; Lehm, lehmizidines; Epiquinamide, epiquinamide; 2-Pyr, 2-substituted pyrrolidine; 3-Pyr, 3-substituted pyrrolidine; 2,5-Pyr, 2,5-disubstituted pyrrolidines; Pyr, pyrrolizidine of indeterminate substitution; 2,6-Pip, 2,6-disubstituted piperidines; Pip, other piperidines; Pyri, pyridines (including epibatidine); GTX, gephyrotoxins; Tricyclic, coccinelline-like tricyclics; SpiroP, spiropyrrolizidines; Necine, unspecified necine base; Unclass, unclassified alkaloids without known structures.

mites. Other data support this general pattern: ants and mites respectively constituted up to 51% and 60% of the stomach contents of the undefended dendrobatids *Allobates talamancae* (*Mebs et al., 2018*) and *Hyloxalus sauli* (*Darst et al., 2005*). Ants and mites compose nearly 50% of the arthropods (36% and 10%, respectively) found in the *Silverstoneia flotator* stomachs we analyzed (*Supplementary file 1*). Sympatric populations of the undefended *Hyloxalus awa* and defended *Epipedobates espinosai* (formerly *E. darwinwallacei* *López-Hervas et al., 2024*) are both diet specialized, with the former consuming mostly ants and beetles and the latter consuming mostly mites and springtails (*Sánchez-Loja et al., 2024*). In a lab experiment, the defended species *Dendrobates tinctorius* preferred fruit fly larvae over ants when given the choice (*Moskowitz et al., 2022b*), suggesting that even in defended species, consumption of possible alkaloid-containing prey is not necessarily a preference. Another study revealed that *Oophaga sylvatica* alkaloid quantity is inversely correlated with numbers of consumed ants and mites; however, this species consumed more mites and ants than sympatric *Hyloxalus elachyhistus* (*Moskowitz et al., 2022a*). The few bufonids that we assessed also show a high proportion of ants and mites in their diet (*Figure 2*). Thus, if we assume that many ants and mites contain alkaloids, it is likely that most if not all dendrobatids and their most recent common ancestors have long been exposed to toxins through their diet.

Few if any experiments have been done to quantify the relationship between natural toxin exposure and toxin resistance in poison frogs. Given the broad diversity of alkaloid classes found in poison frogs (*Daly et al., 2005*), it is very difficult to predict or quantify all possible types or variations of alkaloid resistance that exist across species, or in their ancestors. In animals, the general mechanisms of toxin resistance are avoidance, metabolism, and target modification (*Tarvin et al., 2023*). If an animal does not or cannot *avoid* toxin exposure, it will need to *survive* exposure using toxin metabolism or target modification mechanisms such as biotransformation, elimination, alternative targets, and target-site resistance (see *Tarvin et al., 2023* for more details). Given their diet, dendrobatids clearly do not completely avoid toxin exposure, and thus they are likely to survive exposure using some manner of toxin metabolism or target modification. Indeed, target-site resistance to some alkaloids evolved in several defended dendrobatid clades and in some undefended species (*Tarvin et al., 2017*; *Tarvin et al., 2016*). A few defended species have alternative target mechanisms including binding proteins like alpha-binding globulin (*Alvarez-Buylla et al., 2023*) and saxiphilin (*Abderemane-Ali et al., 2021*) that might prevent alkaloids from accessing their molecular targets (e.g. ion channels). Other mechanisms may also exist. For example, poison frogs may biotransform alkaloids into less toxic forms until they can be eliminated from the body, for example using cytochrome p450s (*Caty et al., 2019*). The mechanism of resistance employed might differ between undefended and defended species, but more research is necessary to understand these patterns.

Although more data are necessary to understand the evolution of toxin resistance in dendrobatids (*Coleman and Cannatella, 2024*), existing data suggest that all or nearly all dendrobatids are exposed to alkaloids (*Figure 2*) and that alkaloid resistance varies among lineages.

## Phases 3 and 4: Evidence for passive accumulation and sequestration in poison frogs

To understand the major evolutionary transition from consuming to sequestering toxins, it is essential to characterize the metabolism and sequestration of alkaloids in defended and undefended dendrobatid lineages (*Gonzalez and Carazzone, 2023*). However, many of the undefended lineages have not been carefully evaluated for the presence or absence of chemical defense. By reviewing existing data, we found that only 31 of the 245 inconspicuous poison frog species described to date *AmphibiaWeb, 2023* have been assessed for toxins, sometimes using methods that would not necessarily detect lipophilic alkaloids (*Supplementary file 2*). Further, prior studies have sometimes misinterpreted or not fully incorporated these data (*Supplementary file 2*, and see below). Our review and reassessment of these studies suggest that at least 11 undefended species might have lipophilic alkaloids: *Allobates femoralis*, *Allobates kingsburyi*, *Allobates zaparo*, *Colostethus ucumari*, *H. elachyhistus*, *Hyloxalus nexipus*, *Hyloxalus vertebralis*, *Hyloxalus yasuni*, *Leucostethus fugax*, *Paruwrobates erythromos*, and *Silverstoneia punctiventris* (*Daly et al., 1987*; *Darst et al., 2005*; *Gonzalez et al., 2021*; *Grant, 2007*; *Moskowitz et al., 2022a*; *Santos and Cannatella, 2011*).

We tested for possible alkaloid presence in additional aposematic and inconspicuously colored poison-frog lineages. Using Gas-Chromatography Mass-Spectrometry (GC-MS), we surveyed 89 animals representing 30 species of Neotropical frogs including 27 dendrobatid species, 1 leptodactylid, and 2 bufonids (*Figure 2*). We also performed a highly sensitive, untargeted analysis —ultrahigh-performance liquid-chromatography heated-electrospray-ionization tandem mass spectrometry (UHPLC-HESI-MS/MS) — of a dendrobatid from an undefended clade (*S. flotator*; 12 individuals) and a species of eleutherodactylid (*Eleutherodactylus cystignathoides*; 3 individuals), in which alkaloid diversities and types, but not quantities, were assessed. Each of the major undefended clades in Dendrobatidae (*Figure 1*, *Table 1*) is represented in our dataset with a total of 13 undefended dendrobatid species surveyed with GC-MS and 1 undefended dendrobatid species surveyed with UHPLC-HESI-MS/MS. As far as we are aware, we provide alkaloid data for the first time for six undefended dendrobatid species (*Rheobates palmatus*, *Allobates juanii*, *Hyloxalus shuar*, *Hyloxalus* sp. Agua Azul, *Silverstoneia* aff. *gutturalis*, and *Silverstoneia erasmios*) and one defended species (*Epipedobates currulao*). We also provide the first alkaloid data for the non-dendrobatids *Amazophrynella siona*, *El. cystignathoides*, and *Lithodytes lineatus* (but see *de Lima Barros et al., 2016*). Because chemical standards for most poison frog alkaloids do not exist, it is not possible to provide absolute quantification of alkaloids. Reported values for GC-MS data are in units of integrated area, which do not directly correspond to alkaloid quantity because of differences in ion yield. Nevertheless, qualitative comparisons of integrated areas can provide insight into how species differ in degrees of magnitude.

Overall, we detected alkaloids in skins from 13 of the 14 undefended dendrobatid species included in our study, although often with less diversity and relatively lower quantities than in defended lineages (*Figure 2*, *Table 1*, *Supplementary file 3*, *Supplementary file 4*). The pervasiveness of low alkaloid levels in undefended dendrobatid lineages (Aromobatinae, Hyloxalinae, some species of Colostethinae) contrasts with the mixed or opposing evidence from previous analyses (*Supplementary file 2*). In addition, our GC-MS assessment revealed substantially higher diversities of alkaloids in defended dendrobatid species than previously reported (*Cipriani and Rivera, 2009*; *Daly et al., 1987*; *Lawrence et al., 2023*; *Moskowitz et al., 2022a,*), and expands knowledge on major classes of alkaloids within genera.

The large number of structures that we identified is in part due to the way we reviewed GC-MS data: in addition to searching for alkaloids with known fragmentation patterns, we also searched for anything that could qualify as an alkaloid mass spectrometrically but that may not match a previously known structure in a reference database. Similarly, the analysis of UHPLC-HESI-MS/MS data was untargeted, and thus enables a broader survey of chemistry compared to that from prior GC-MS studies. Structural annotations in our UHPLC-HESI-MS/MS analysis were made using CANOPUS, a deep neural network that is able to classify unknown metabolites based on MS/MS fragmentation patterns, with 99.7% accuracy in cross-validation (*Dührkop et al., 2021*).

Although contamination across samples is possible, it is unlikely to invalidate the identification of alkaloids in undefended species based on the following. (1) At several sites, we only sampled undefended species, and these individuals were found to contain alkaloids (e.g. Las Brisas: *R. palmatus*; El Valle: *S.* aff. *gutturalis*; Santa Maria: *H.* sp. Agua Azul); that is these cannot possibly have come

from contamination by defended species. (2) At one site where we collected both undefended and defended species, the undefended species shows no alkaloids (Lita: *Hyloxalus toachi*); i.e., the preparation of both types does not imply cross-contamination of samples. (3) At two sites where the undefended species were prepared on a different day from the defended species (Valle Hermoso: *H. awa* and *Epipedobates boulengeri*; Canelos: *L. fugax* and *Ameerega hahneli*) and could not have been cross-contaminated, the undefended species still show evidence of alkaloids. (4) All chromatograms in the GC-MS sequence and integration data were inspected manually. Peaks with low areas or following samples with high areas and subject to carryover were excluded from further analysis. (5) Data collected by a different team and analyzed with different methods also identify alkaloids in an undefended dendrobatid (*S. flotator*) from Panama.

## Aromobatinae

For Aromobatinae, we surveyed the undefended genera *Rheobates* and *Allobates*. Alkaloids were detected in all four *R. palmatus* individuals sampled, with one individual having at least four classes of compounds represented (4,6-disubstituted quinolizidines, 3,5-disubstituted indolizidines, 3,5-disubstituted pyrrolizidines, and unclassified). We found that five species of *Allobates* all had detectable levels of alkaloids. *Allobates insperatus* had a relatively high level of alkaloid diversity, with at least 18 alkaloids from nine classes detected, and at least one class found in each of the eight sampled individuals. In contrast, only one unclassified alkaloid was identified in a single individual of *Al. juanii* while two were found in one individual of *Al. kingsburyi*. At least two alkaloids were identified in each of the three sampled individuals of *Al. talamancae* (including the lehmizidine **277** A and five new alkaloids). Eight alkaloids were identified in the single surveyed *Al. zaparo* individual (including the spiropyrrolizidines **222–1** and **222–2** as well as six unclassified alkaloids). Prior assessments using thin-layer chromatography suggested the presence of alkaloids in three *Al. kingsburyi* (*Santos and Cannatella, 2011*), but none in 12 *Al. insperatus* (*Darst et al., 2005*). Four studies (*Supplementary file 2*) failed to identify any alkaloids in *Al. talamancae*. *Allobates zaparo* was shown to possibly have trace alkaloids, although the interpretation of these data was absence of alkaloids (*Darst et al., 2005*). There are no known defended species from this subfamily, although we note conflicting evidence for the presence of alkaloids in *Al. femoralis* (*Amézquita et al., 2017*; *Daly et al., 1987*; *Sanchez et al., 2019*; *Saporito and Grant, 2018*; *Supplementary file 2*).

## Colostethinae

Within Colostethinae, we surveyed individuals from two undefended clades, *Leucostethus* and *Silverstoneia*, and from two defended clades, *Epipedobates* and *Ameerega*. From *L. fugax*, we identified a total of twelve 5-substituted indolizidine, 5,6-disubstituted indolizidine, pyrrolidine, spiropyrrolizidine, and unclassified alkaloids (**196** A, **225** C, **222–1**, **222–2**, and eight new alkaloids), with three to eight unique compounds detected in each of the eight sampled individuals. Our data are consistent with prior thin-layer chromatography data showing that *L. fugax* tested positive for skin compounds (*Santos and Cannatella, 2011*), although prior interpretations of these data were different (*Supplementary file 2*). We also surveyed two species of *Silverstoneia* with GC-MS. We found alkaloids in all nine *S.* aff. *gutturalis*, with a total of 14 alkaloids identified across seven classes (**196** A, **223I**, **233** A, **235B**, **237** U, three isomers of **239AB**, two isomers of **239** CD, and four new alkaloids). In just two individuals of *S. erasmios,* we detected a total of 26 alkaloids, including some pumiliotoxins (**325B**, **323B**) and pyrrolizidines (**225** C). *Silverstoneia erasmios* and *S.* aff. *gutturalis* had not been surveyed for alkaloids previously, but thirteen alkaloids were found in eight individuals of a congener (*S. punctiventris*; *Gonzalez et al., 2021*). In addition, in our more conservative UHPLC-HESI-MS/MS analysis of *S. flotator*, from which we only report compounds with formulae or from classes previously known for lipophilic alkaloids of frogs (*Daly et al., 2005*), we identified the presence of alkaloids in 5 of 12 sampled individuals (a quinolizidine and epibatidine; *Supplementary file 5*, *Supplementary file 6*). When we expand our analysis to include any compound assigned to the 'alkaloid pathway' by NPClassifier (>99% alkaloid pathway probability; *Supplementary file 5*), we identified a total of 67 compounds, some of which were present in each individual (*Supplementary file 5*). Although the assignments made by this pipeline are broad and include diverse nitrogen-containing metabolites such as biogenic amines (*Supplementary file 6*), it is possible that some represent additional

lipophilic alkaloids whose structures and formulae are undescribed. Note that UHPLC-HESI-MS/MS data should not be directly compared to GC-MS data (see *Table 1* legend).

In terms of the defended clades of Colostethinae that we sampled, most of the individual skins of *Epipedobates* and *Ameerega* contained dozens to more than one hundred unique alkaloids (see *Supplementary file 4* for full details). For *Ameerega*, we surveyed five individuals representing two species, all of which had integrated areas that were more than 75,000 x greater compared to individuals of its sister clade, *Leucostethus* (*Table 1*). Similarly, alkaloid diversity was 10–20 x greater in *Ameerega* than in *Leucostethus*. Histrionicotoxins and decahydroquinolines were considered previously to be the dominant alkaloid classes in genus *Ameerega Daly et al., 2009*; here we also found high levels of indolizidines (*Figure 2*). Patterns for *Epipedobates* as compared to sister genus *Silverstoneia* were similar, although less extreme. We surveyed 13 individuals representing seven species in *Epipedobates* and identified at least 370 alkaloids, which contrasts with studies using a less sensitive method (thin-layer chromatography) that found mixed evidence for the presence of alkaloids in *E.* aff. *espinosai* (then referred to as *E. boulengeri*) and *E. machalilla* (*Darst et al., 2005*; *Santos and Cannatella, 2011*). However, the quantity and diversity of alkaloids in *Epipedobates machalilla* was substantially lower than in other *Epipedobates* species, occurring at levels similar to *Silverstoneia* spp. (*Table 1*, *Figure 2*). Except for *E. machalilla*, each *Epipedobates* species had about 10 x greater quantities and diversities of alkaloids compared to members of *Silverstoneia*. We found trace levels of epibatidine in *Epipedobates anthonyi* but not in other *Epipedobates* species. Epibatidines have also been detected in *E. espinosai, Ameerega silverstonei, S. flotator* (*Daly et al., 1999*; this study), and *Ameerega petersi* or a closely related, undescribed species (reported as *Dendrobates pictus* from Loreto, Peru in *Daly et al., 1987*, but see taxonomic revision by *Guillory et al., 2020*).

## Hyloxalinae

Hyloxalinae is generally considered an undefended clade (*Supplementary file 2*). We surveyed four species of *Hyloxalus*, three of which had detectable levels of alkaloids. We identified 17 different alkaloids in *H. awa* (**197D**, **197** H, **199B**, **217B**, **221** P, **223AB**, **231** A, **231** C, **247E**, and eight previously undescribed alkaloids), with the seven sampled individuals having 0–12 alkaloids each. We detected five alkaloids in a single individual of *H. shuar* (**197D**, **199B**, **237** G, and two isomers of **239** K) and eight alkaloids in a single individual of *H.* sp. Agua Azul (**195** C, **197D**, **199B**, **251** K, and four new alkaloids). Our detection of low levels of alkaloids in *H. awa* is consistent with the observations that avian predators consume *H. awa* (*Darst and Cummings, 2006*). No alkaloids were detected in two individuals of *H. toachi*, the only undefended dendrobatid species from which we failed to detect alkaloids. Previously, a GC-MS assessment revealed that *P. erythromos* contained 5,8-disubstituted indolizidine **251B**, allopumiliotoxin **267** A, and unclassified alkaloid **281D** (*Daly et al., 1987*). *Hyloxalus azureiventris* is also thought be able to accumulate alkaloids (*Daly, 1998*; *Saporito et al., 2009*) and thin-layer chromatography suggested the presence of alkaloids in two assessed *H. yasuni* (previously identified as *Hyloxalus maculosus*), one of three *H. nexipus*, and two of five *H. vertebralis* (*Santos and Cannatella, 2011*), although prior interpretation of these data differed (*Supplementary file 2*). Our data support the widespread presence of low levels of alkaloids in this group.

## Dendrobatinae

According to the most recent phylogenetic reconstructions (*Santos et al., 2014*), the sister clade to Hyloxalinae is Dendrobatinae (*Figure 1*). Dendrobatinae contains exclusively (or near exclusively) toxic species. From this subfamily, we surveyed 18 individuals representing five species using GC-MS. We identified a total of 187 unique alkaloids from four *Phyllobates aurotaenia*, 316 alkaloids from five *O. sylvatica*, and 213 alkaloids from three *Dendrobates truncatus*. These three species are all relatively large poison frogs (snout-vent lengths 20–35 mm; *Supplementary file 3*), which may in part explain their high alkaloid diversities and quantities (*Jeckel et al., 2015*; *Saporito et al., 2010*). In *Andinobates minutus* and *Andinobates fulguritus*, which are members of the same subfamily but are much smaller in size (11–15 mm; *Supplementary file 1*, *Supplementary file 3*), we detected 129 and 109 alkaloids, respectively. Three of the *An. minutus* individuals were juveniles. The total alkaloid quantities (integrated areas) in *D. truncatus* and *O. sylvatica* were comparable to those of *Ameerega* but were higher than quantities detected in *Epipedobates*. We also report for the first time, to the best of our knowledge, the occurrence of N-methyldecahydroquinolines outside of the genera *Adelphobates*,

*Ameerega*, *Dendrobates*, *Oophaga*, and *Ranitomeya* (in *E.* aff. *espinosai*, *E. currulao*, *S.* aff. *gutturalis*, *An. minutus*, *An. fulguritus*, *P. aurotaenia*, and *Al. insperatus*; *Daly et al., 2009*; *Hovey et al., 2018*; *Jeckel et al., 2019*; *Lawrence et al., 2019*; *Stuckert et al., 2014*). The ability to N-methylate deca-hydroquinoline (demonstrated experimentally in *Adelphobates galactonotus*, *Dendorbates auratus*, and *Ranitomeya ventrimaculata Jeckel, 2021*; *Jeckel et al., 2022*) may thus be conserved in dendro-batids, or, non-exclusively, arthropod sources of the alkaloid class (likely myrmicine ants *Jones et al., 1999*) are widespread.

## Other frog families

Outside of Dendrobatidae, we detected a new unclassified alkaloid, New159, in each of two *A. siona* (Bufonidae) and four alkaloids in one individual of *Atelopus* aff. *spurrelli* (Bufonidae; 3,5-disubstituted pyrrolizidine **237** R-1, decahydroquinoline **243** A-3, 5,8-disubstituted indolizidine **251B**-2, and an unclassified alkaloid, New267-2). As far as we know, the detection of a decahydroquinoline and a 3,5-disubstituted pyrrolizidine in a bufonid frog other than *Melanophryniscus* (*Rodríguez et al., 2017*) is novel and may provide useful context for understanding the evolution of chemical defense in the family. We detected no alkaloids in two *Li. lineatus* (Leptodactylidae) individuals, which is surprising because *Li. lineatus* has been hypothesized to be a Müllerian mimic of poison frogs, although the composition of its chemical defenses may be primarily proteinaceous (*Prates et al., 2012*). These findings are also interesting in light of the fact that *Li. lineatus* live and breed in ant colonies using chemical signals that provide camouflage (*de Lima Barros et al., 2016*). In addition, while we recov-ered no alkaloids in three sampled individuals of the frog *El. cystignathoides* (Eleutherodactylidae) with UHPLC-HESI-MS/MS when we applied our stringent search criteria, we identified 55 metabolites assigned to the alkaloid pathway at >99% probability. Forty of these appear to be identical to those identified in *S. flotator* according to our analyses (*Supplementary file 5*, *Supplementary file 6*). Some of these could be widespread byproducts of frog metabolism (or symbiont metabolism). A few species of *Eleutherodactylus* frogs from Cuba are thought to sequester alkaloids (*Rodríguez et al., 2013*) and alkaloid sequestration evolved in the bufonid genus *Melanophryniscus* (*Daly et al., 1984*; *Hantak et al., 2013*). The presence of low levels of alkaloids in other (non-sequestering) species of Bufonidae and the possibility of some exogenous but as of yet undescribed alkaloids in *El. cystignathoides* reflect that passive accumulation may have evolved in an older ancestor shared by the three families, predating convergent origins of sequestration in all three groups.

## Predictions arising from the passive-accumulation hypothesis

Data from this and other studies point to the ubiquity of mites and ants in nearly all dendrobatid diets, and possibly more generally in other leaf-litter dwelling frogs (*Figure 2*). This finding in concert with the detection of low levels of alkaloids in the lineages that putatively lack chemical defenses leads us to hypothesize that dietary shifts are not sufficient to explain the presence or absence of the chemical defense phenotype within Dendrobatidae or possibly in other families (Bufonidae). The total amount of alkaloids accumulated is a result of multiple processes including toxin intake, elimination, and sequestration (*Figure 3*) — not just intake alone.

For example, dendrobatid species vary in their ability to eliminate alkaloids. Some appear to lack specific transport and storage mechanisms for consumed alkaloids ('sequestration'), yet they have detectable levels of alkaloids in their skin; we refer to this phenotype as passive accumulation and suggest that it is an evolutionary intermediate between toxin consumption (with no sequestration) and sequestration (*Figure 3*). We predict that the ancestral state of poison frogs and potentially other clades with alkaloid-sequestering species (e.g. Bufonidae: *Melanophryniscus*, Eleutherodactyl-idae: *Eleutherodactylus*, and Mantellidae: *Mantella*) is alkaloid consumption and low levels of alkaloid resistance, accompanied by passive alkaloid accumulation (e.g. see *Figures 1 and 3*). Alternatively, passive accumulation may have arisen in an even earlier ancestor. That we detected alkaloids in two genera of bufonid toads could reflect a single origin of passive accumulation in the ancestor of the clade that includes Dendrobatidae and its sister group (the clade comprised of the Terranana [including Eleutherodactylidae], Bufonidae, Leptodactylidae, and Odontophrynidae; *AmphibiaWeb, 2023*; *Blackburn and Wake, 2011*; *Feng et al., 2017*; *Streicher et al., 2018*; *Yuan et al., 2019*). Further sampling for alkaloids within Eleutherodactylidae and Leptodactylidae could reveal whether passive accumulation has persisted in these clades. Discriminating a single origin — no matter the

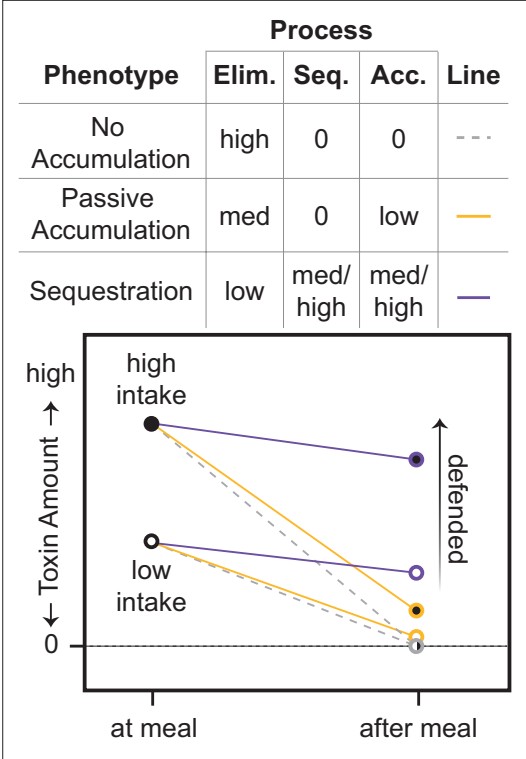

**Figure 3.** Hypothesized physiological processes that interact to determine the defense phenotype: toxin intake, toxin elimination (Elim.), and toxin sequestration (Seq.). A new paradigm: the passive-accumulation hypothesis for definitions. Although toxin intake sets a maximum for the total possible amount of toxin accumulation (Acc.), it cannot fully explain the defensive phenotype. We hypothesize that an undefended "no accumulation" phenotype is characterized by the absence of any ability to sequester toxins in combination with a high rate of elimination, resulting in 0 toxin accumulation (dashed gray lines); this phenotype is a likely ancestral state for many animals. In contrast, we hypothesize that an undefended passive-accumulation phenotype is characterized by lower elimination than the no accumulation phenotype, leading to a low amount of toxin accumulation (yellow lines). We hypothesize that a defended sequestration phenotype evolves from an intermediate passive-accumulation phenotype through the addition of novel sequestration mechanisms, and possibly even lower elimination rates, that result in high toxin accumulation and the defended phenotype (purple lines).

timing — from multiple ones would require more extensive alkaloid surveys, as we only assessed four non-dendrobatid species.

Here, we propose and discuss three additional predictions arising from the passive-accumulation hypothesis that would help further evaluate the validity of a four-phase model.

## Prediction (1) We predict that some toxin resistance evolves prior to or in concert with passive accumulation, and that it increases or changes once sequestration mechanisms evolve

Alkaloid resistance is associated with alkaloid sequestration in dendrobatid poison frogs (*Tarvin et al., 2017*; *Tarvin et al., 2016*). We anticipate that some alkaloid resistance evolved in the ancestor of Dendrobatidae or in an even older ancestor, but is yet to be described (*Darst et al., 2005*; *Santos et al., 2016*; *Figure 1*). Such resistance may be difficult to characterize using the comparative method if it involves mutations of small effect (*Ffrench-Constant et al., 2004*), pleiotropic processes, or undescribed physiological adaptations (e.g. *Alvarez-Buylla et al., 2023*). Regardless, it appears that arthropods likely to contain alkaloids are widespread among the regular diets of defended and undefended dendrobatid poison frogs (*Figure 2*; *Darst et al., 2005*; *Santos and Cannatella, 2011*; *Toft, 1995*). Short-term alkaloid feeding experiments e.g. *Daly et al., 1994b*; *Sanchez et al., 2019* demonstrate that both defended and undefended frogs can survive the immediate effects of alkaloid intake, although the degree of resistance and the alkaloids that different species can resist vary. An experiment conducted by *Abderemane-Ali et al., 2021* showed that both aposematic and (presumably) undefended frogs can withstand several highly toxic alkaloids in quantities greater than what the frogs are likely to experience in nature. Two aposematic dendrobatids (*D. tinctorius* and *Phyllobates terribilis*), an aposematic *Mantella* (*Mantella aurantiaca*), and the putatively undefended rhacophorid *Polypedates leucomystax* (a congener is reported to contain TTX *Tanu et al., 2001*) — recovered from injections of the highly toxic alkaloids batrachotoxin and tetrodotoxin delivered at 20 x the mouse $LD_{50}$ value.

The three aposematic species also survived a third potent alkaloid (saxitoxin), but *P. leucomystax* did not. Other work revealed no signs of intoxication in two undefended hylids (*Hyla cinerea* and *Boana bandeirantes*) after 2 wk of oral administration of histrionicotoxin **235** A and decahydroquinoline (*Jeckel, 2021*). In an epibatidine-feeding experiment with five aposematic dendrobatid species (*E. anthonyi, Ranitomeya variabilis, Ranitomeya imitator, Phyllobates vittatus,* and *D. tinctorius*), *Waters*

*et al., 2024* found that *E. anthonyi* was adversely affected by the initial dose of epibatidine, reflecting either a body-size effect or species-level variation in epibatidine resistance.

Different types of resistance may be important during different evolutionary phases leading to chemical defense. For example, many mechanisms of toxin metabolism are common to all animals and were likely used by the ancestors of most if not all animals that eventually evolved toxin sequestration, including poison frogs (*Tarvin et al., 2023*). Although one might expect that toxin metabolism may also prevent toxin sequestration, the ability to metabolize toxins can in some cases augment toxin defenses (*Douglas et al., 2022*), increase the toxicity of a compound (e.g. pumiliotoxin to allopumiliotoxin in the poison frogs *Ad. galactonotus, Adelphobates castaneoticus, D. auratus, D. tinctorius,* and *R. ventrimaculata Alvarez-Buylla et al., 2022*; *Daly et al., 2003*; *Jeckel, 2021*), or result in some amount of passive accumulation through increased toxin exposure (*Douglas et al., 2022*; *Karageorgi et al., 2019*). If toxin intake increases or is sustained over long evolutionary periods, selection may favor other mechanisms of resistance, such as target-site resistance, which can eliminate the cost of toxin exposure by making the targeted protein insensitive to the toxin (*Tarvin et al., 2017*). Indeed, other toxin-sequestering animals often have specialized mechanisms of toxin resistance when compared to toxin-free relatives (*Tarvin et al., 2023*). For example, three amino acid replacements in the ATPα protein evolved in association with cardenolide sequestration in Danainae butterflies (*Karageorgi et al., 2019*; *Petschenka et al., 2013*) and predatory fireflies that sequester lucibufagins have ATPα gene duplications that enhance lucibufagin resistance (*Yang et al., 2023*).

These data suggest that some dendrobatids and other frog species have a minimal level of resistance to alkaloids, yet more data from undefended frogs will be necessary to reconstruct the evolutionary history of the trait.

## Prediction (2) We predict that in species with passive accumulation the rate of toxin elimination is slower than in those with no accumulation and faster than in those with sequestration

Only a few studies have reviewed toxin metabolism and elimination (clearance from the body) in dendrobatids. One study demonstrated that the undefended *Al. femoralis* and undefended hylid *Hy. cinerea* accumulated less than 1% of orally administered alkaloids into the skin, yet the alkaloids were absent (or present in only trace amounts) in the feces (*Jeckel, 2021*). In the same experiment, the defended dendrobatids *Ad. galactonotus* and *D. tinctorius* efficiently sequestered the alkaloids, with only trace quantities detected in the feces. These results hint at an unknown but possibly conserved mechanism for metabolism of alkaloids in anurans. Even among defended dendrobatids, there appears to be species-level variation and plasticity in the metabolism and elimination of alkaloids. *Epipedobates anthonyi, Ra. variabilis,* and *Ra. imitator* accumulate more than twice as much ingested epibatidine compared to *P. vittatus* and *D. tinctorius* (*Waters et al., 2024*). *Oophaga sylvatica* and *D. tinctorius* upregulate detoxification genes such as cytochrome p450s upon alkaloid consumption (*Alvarez-Buylla et al., 2022*; *Caty et al., 2019*). *Adelphobates galactonotus* sequesters the alkaloids histrionicotoxin **235** A and decahydroquinoline less efficiently at higher doses (*Jeckel et al., 2022*). Some species metabolically alter the structure of alkaloids: *Ad. galactonotus, Ad. castaneoticus, D. auratus, D. tinctorius,* and *Ra. ventrimaculata* can hydroxylate pumiliotoxin **251D** (*Alvarez-Buylla et al., 2022*; *Daly et al., 2003*; *Jeckel, 2021*), making it more toxic (to mice); *Ad. galactonotus, D. auratus,* and *Ra. ventrimaculata* can also N-methylate decahydroquinoline (*Jeckel, 2021*; *Jeckel et al., 2022*). These studies indicate that alkaloid elimination rate and metabolism vary among defended species, but not enough information exists to infer much about elimination rates in undefended lineages with or without passive accumulation. Given the experimental demonstration of less efficient alkaloid uptake in undefended frogs — in combination with our data that show that despite likely ingesting alkaloid-bearing prey regularly in the wild, undefended frogs show much lower levels of alkaloids in the skin (*Figure 2*) — we hypothesize that toxin elimination rates in undefended lineages are faster or more efficient than rates in defended lineages and are slower than lineages with no accumulation (e.g. *Figure 3*). More nuanced versions of this model could also be envisioned. For example, elimination rates in defended species could still modulate the amount of toxins ultimately accumulated, with lower elimination rates resulting in a higher proportion of toxin accumulation overall. Additional data are necessary regarding toxicokinetics of consumed alkaloids in several tissues.

## Prediction (3) We predict that sequestration mechanisms are absent in undefended lineages

*Daly, 1998*; *Daly et al., 1994b* hypothesized that there was an alkaloid uptake system present in the ancestor of Dendrobatidae that is "overexpressed" in the defended lineages. This hypothesis remains to be tested. Our model posits that sequestration mechanisms (*Figures 1 and 3*) are unique to chemically defended species. Alternatively, if the mechanisms of toxin transport and/or storage exist in undefended species, they seem to be expressed at such a low level that they only result in a trace level of toxin accumulation. In order to distinguish between these two possibilities, we first will need to better understand the molecular mechanisms underlying toxin sequestration.

To date, little is known regarding the mechanisms of toxin sequestration in poison frogs or in other toxin-sequestering animals. An alkaloid-binding globulin was recently characterized in the poison frog *O. sylvatica* (*Alvarez-Buylla et al., 2023*). While plasma assays demonstrated that the defended species *O. sylvatica*, *Epipedobates tricolor*, and *D. tinctorius* can bind and sequester a pumiliotoxin-like photoprobe, plasma from the undefended *Al. femoralis* showed no binding activity. In addition, the evolutionarily distant mantellid species *M. aurantiaca*, which sequesters alkaloids, did not show binding activity. These data hint at variation in molecular mechanisms for alkaloid uptake across lineages, which may be tuned to availability of specific alkaloids in each species' diet.

The potential absence of sequestration mechanisms in the undefended *Al. femoralis* are consistent not only with our alkaloid data from wild-caught frogs, but also with experimental data. Using GC-MS, researchers did not detect any alkaloids in the skins of two undefended dendrobatids (*Al. talamancae* and *Colostethus panamansis*) after the frogs consumed fruit flies dusted with 5,8-disubstituted indolizidine **209B**, decahydroquinoline **195** A, and histrionicotoxin **285** C for five weeks (*Daly et al., 1994b*). Other unpublished data suggest that the brightly colored but undefended *H. azureiventris* are unable to accumulate alkaloids from fruit flies (the sample size and alkaloid identities are unknown), although *H. azureiventris* apparently accumulated four distinct alkaloids from a methanol-saline solution (*Saporito et al., 2009*). After oral administration of decahydroquinoline and histrionicotoxin **235** A, the undefended hylid *Hyla cinerea* cleared almost all consumed alkaloids (accumulating between 0.01 to 0.1%), the undefended *Al. femoralis* accumulated only trace amounts of decahydroquinoline (~1%), and the defended *Ad. galactonotus* and *D. tinctorius* sequestered on average ~10% (decahydroquinoline) or ~50% (histrionicotoxin **235** A) (*Jeckel, 2021*). Sparteine, a quinolizidine structurally similar to the common 'izidine' alkaloids in poison frogs, was detected in the skin of a single *Al. femoralis* individual after the frog was fed sparteine-dusted fruit flies for over a month, but the experimental methods prohibited quantification of the alkaloid (*Sanchez et al., 2019*).

Additional data on potential uptake mechanisms in dendrobatids exist for benzocaine, a synthetic lipophilic compound that is used for anesthesia and euthanasia in amphibians. Benzocaine is readily taken up orally to the skin in the defended poison frog *D. auratus*, the undefended ranid (*Lithobates clamitans*), and the alkaloid-sequestering bufonid *Melanophryniscus moreirae* (*Saporito and Grant, 2018*). Although the same amount of benzocaine was injected into each frog, twice as much benzocaine was recovered from *D. auratus* than *Li. clamitans* and three times as much was recovered from *Me. moreirae* (see their Fig. 2), suggesting that lipophilic compound uptake occurs without specialized mechanisms of sequestration in *Li. clamitans* (e.g. possibly passive accumulation) but that *D. auratus* and *Me. moreirae* likely have distinct sequestration mechanisms that result in much higher levels of benzocaine accumulation.

In contrast to sequestration, passive accumulation would be expected to result in the diffusion of alkaloids across many tissues, rather than concentration of alkaloids within a specific tissue. Desorption electrospray ionization mass spectrometry imaging data indicate that alkaloids diffuse across various tissues in the defended dendrobatid *D. tinctorius* immediately following intake, possibly an evolutionary trace of the low elimination rates that may have initially evolved in an ancestor with the passive accumulation phenotype (*Jeckel et al., 2020*). It would be beneficial to conduct a time-series study to show how tissue-specific accumulation patterns change after feeding in different species. Clearly, more data will be necessary to evaluate phylogenetic patterns and mechanisms of sequestration, and to test the hypotheses presented here regarding passive accumulation as an intermediate evolutionary phase.

## Other factors that may shape the evolution of acquired chemical defenses

Many animals occasionally or frequently consume toxins, and a multitude have evolved toxin resistance. Some invertebrate pests resist pesticides (*Andreev et al., 1999*; *Chiu et al., 2008*; *Daborn et al., 2002*; *Ffrench-Constant, 2013*), many insect herbivores resist plant toxins (*Agrawal et al., 2012*; *Dobler et al., 2011*), some predators resist toxic prey (*Arbuckle et al., 2017*), and many animals resist environmental pollutants (*Whitehead et al., 2017*). Our model predicts that some or many of these may be on their way towards evolving acquired chemical defenses. Yet, not all toxin-exposed or toxin-resistant species inevitably evolve chemical defenses, presumably because the ecological context or physiology that favors accumulation is absent or because resisting and accumulating toxins is too costly.

Acquired chemical defenses usually evolve within the context of a tri-trophic interaction: animals in the middle of the food web accumulate toxins from their prey, and possible predators or parasites are deterred by the accumulated toxin (*Agrawal, 2000*). This phenomenon is referred to as enemy-free space, i.e., escape from parasitism or predation (*Jeffries and Lawton, 1984*). If there is no predator or parasite present to exert selection on a toxin-consuming animal, there may be no benefit for the animal to accumulate the toxins. Furthermore, some chemicals may not be able to be accumulated because of how they interact with the physiology of an animal (e.g. *Mebs et al., 2016*). Thus, the evolution of chemical defenses may be constrained by the specific chemicals present in an ecosystem, the existing trophic interactions among species, and the physiology of predators and parasites in relation to the chemicals in question.

Origins of chemical defenses are also shaped by the cost of resisting and accumulating toxins, which can change over evolutionary time as animals adapt to novel relationships with toxins. In poison frogs and other toxin-accumulating animals, it is common to observe a few amino acid substitutions in ion channels that provide target-site resistance to alkaloids but adversely affect the function of the protein; these substitutions are often accompanied by additional, compensatory substitutions that restore protein function without affecting resistance (*Karageorgi et al., 2019*; *Mohammadi et al., 2021*; *Reid et al., 2016*; *Tarvin et al., 2017*; *Zhang et al., 2016*). It is rare but possible to observe species that lack (known) compensatory substitutions (*Tarvin et al., 2017*), suggesting that species are under strong selection to overcome some costs of target-site resistance. In one species of garter snake (*Thamnophis sirtalis*), the cost of target-site resistance in a voltage-gated sodium channel is not completely offset as animals with target-site resistance have reduced crawl speeds (*Hague et al., 2018*). In some insects, resistance to insecticides comes with a cost in fecundity or survival (*Kliot and Ghanim, 2012*). For example, the aphid *Aphis nerii* experiences trade-offs between population growth and defense effectiveness (*Züst et al., 2018*). As far as we are aware, the possible lifetime fitness costs (e.g. in reproductive success) of alkaloid consumption in dendrobatids have not been measured.

Once chemical defenses evolve, they are often further shaped by co-evolution between the defended prey and their predators (*Brodie and Brodie, 1990*; *Bucciarelli et al., 2022*), which can result in the appearance of visual or morphological signals, mimicry, and even the loss of defenses in the prey if the predator evolves sufficient resistance (*Brodie and Brodie, 1991*; *Brown and Trigo, 1994*; *Crothers et al., 2016*). These additional ecological factors in turn shape the physiology of an animal in ways that may further promote evolutionary innovation (*Loeffler-Henry et al., 2023*; *Przeczek et al., 2008*; *Santos et al., 2014*). In sum, various factors interact in a dynamic equilibrium over short and long timeframes to shape chemical defenses.

## The passive-accumulation phenotype in a broader evolutionary context

Passive accumulation of toxins is not a novel concept, as it has been discussed previously in terms of self-medication (*Clayton and Wolfe, 1993*; *Singer et al., 2009*) and bioaccumulation (e.g. of environmental pollutants; *Butler, 1978*; *Spurgeon et al., 2020*; *Streit, 1992*), and we propose that it is also conceptually analogous to some medical treatments in humans such as chemotherapy. Any organism that consumes something toxic might simultaneously suffer from toxin exposure yet benefit from the compound's effect on disease, infection, parasites, or predators. For example, in the presence of parasitoids, *Drosophila suzukii* flies preferentially lay their eggs on the insecticide atropine, which protects them from being parasitized but prolongs development (*Poyet et al.,*

*2017*). Mechanisms that likely underlie passive accumulation may also be analogous to key organismal functions (*Duffey, 1980*). For example, humans accumulate vitamin E in the liver (*Violet et al., 2020*) and use a transfer protein abundant in liver cells to shuttle the vitamin into the plasma where it becomes bioavailable (*Arita et al., 1995*). The transition from passive accumulation to sequestration in poison frogs may similarly rely on the use of proteins that bind to and transport alkaloids (*Alvarez-Buylla et al., 2023*).

If toxin accumulation is both low-cost and beneficial, slow toxin elimination rates could evolve quite readily, resulting in passive accumulation. Two recent studies support the idea that some toxin resistance permits toxin intake and results in passive accumulation. In one, nicotine-resistant *Drosophila melanogaster* fruit flies that were fed nicotine accumulated measurable amounts of the toxin in their bodies, more than nicotine-sensitive flies (*Douglas et al., 2022*). In another study, ouabain-resistant *D. melanogaster* flies that were fed ouabain accumulated measurable amounts of ouabain in their bodies, more than ouabain-sensitive flies (*Karageorgi et al., 2019*). In a another example, the sawfly *Athalia rosae* shows constant turnover of its glucosinolate toxins, suggesting that these insects cannot effectively store glucosinolates, yet their metabolic clearing is inefficient enough that they still maintain a high level of toxins in the hemolymph (*Müller and Wittstock, 2005*). It is conceivable, then, that in some cases, passive accumulation could result in chemical defense through a mechanism that enables high net toxin intake, followed by evasion of elimination mechanisms, passive entry into the bloodstream, and diffusion into other tissues.

Are these cases of sequestration? Under our definition they are not, given that these species do not actively transport and store these compounds, as far as we know. Rather, these organisms merely fail to efficiently metabolize and eliminate these compounds, leading to their temporary diffusion in tissues and providing a transient benefit against parasites or predators. Evidence for this passive-accumulation phenotype as an intermediate stage on the path towards toxin sequestration is scarce, but passive accumulation is a pervasive pattern in studies of ecological toxicology and may be more common in lineages that evolved toxin sequestration than we currently know.

## Limitations

Our study presents a novel alkaloid dataset for dendrobatid frogs and some relatives, yet it is limited in the following ways. For some species we only sampled one or two individuals, which may paint an incomplete picture of toxin diversity, toxin quantity, and diet in the group. Poison frogs vary substantially over time, space, and seasons in their alkaloid profiles and diets (*Agudelo-Cantero et al., 2015*; *Saporito et al., 2007a*), yet we did not conduct serial sampling over a broad geographic range for each species. Standards are unavailable for most frog alkaloids and thus we could not measure absolute quantity. Relative quantitation of GC-MS data was performed based on integration of the extracted ion chromatogram of the base peak for each alkaloid for maximum sensitivity and selectivity. The nature of these data means that qualitative comparisons may be meaningful but quantitative comparisons across alkaloid structures could be misleading, especially given our small sample sizes for some species. Finally, batrachotoxin and tetrodotoxin are too heavy to study using GC-MS; we cannot exclude the possibility that they occur in the sampled species.

## Conclusion

The large-scale evolutionary transition from consuming to sequestering toxins has occurred in a plethora of invertebrates (*Duffey, 1980*) and vertebrates (*Savitzky et al., 2012*). Here, we provide new evidence showing that undefended poison frogs and frogs in a closely related family (Bufonidae) contain measurable amounts of alkaloids, and we confirm that they consume some amount of toxic arthropod prey. We propose that passive accumulation of consumed alkaloids is an ancestral state in Dendrobatidae, and possibly in related taxa, and that selection acts on toxin elimination and resistance to result in toxin accumulation and chemical defense. Future studies of the toxicokinetics of alkaloids in different tissues of both defended and undefended poison frogs will shed light on these putative intermediate evolutionary steps. In turn, insights from poison frog physiology will provide a novel perspective for the development of human therapeutics, which modulate some of the same pharmacokinetic processes.

## Methods

### Field collection

*Silverstoneia flotator* and *El. cystignathoides* were collected and euthanized with benzocaine in 2022 in Gamboa, Panama (9.1373,–79.723183) and in 2024 in Austin, Texas, USA (30.285,–97.736 and 30.292487,–97.737874), respectively. Dorsal and ventral skins were removed and placed separately in ~1 mL MeOH in 1-dram glass vials for UHPLC-HESI-MS/MS analyses (see below). All other species were collected in 2014 and euthanized with an overdose of lidocaine. Whole skins were removed and placed in ~1 mL MeOH in glass vials with PTFE-lined caps. Stomachs of all species were removed and placed in 95% ethanol. Instruments and dissection surfaces were cleaned with 95% ethanol between dissections. Species were selected with the goal of broad sampling of evolutionary lineages in Dendrobatidae; our protocols followed the ARRIVE guidelines where applicable (*Kilkenny et al., 2010*). The number of individuals sampled per species was opportunistic. For each *Epipedobates* species, a subset of total samples available were randomly selected to be included; the full dataset will be published in another paper focused on variation within the genus.

### Alkaloid identification and quantification

For samples from Ecuador and Colombia, a 100 µL aliquot of the MeOH was sampled from each vial and transferred to a 200 µL limited volume insert and analyzed directly by GC-MS. The system used was a Thermo AS-3000 autosampler interfaced to a Trace GC Ultra interfaced to an iTQ 1100 ion trap mass spectrometer autotuned with FC-43 (PFTBA) operating in positive ion mode. AS conditions were as follows: 2 pre-wash cycles of 5 µL MeOH, then 3 plunger strokes and withdrawal of 1.00 µL sample with 1 µL air gap, injection with no pre- or post-injection dwell followed by 3 post wash cycles of 5 µL MeOH. GC conditions were as follows: splitless injection, splitless time 1.00 min with surge (200 kPa for 0.70 min, to sharpen early peaks), split flow 50 mL/min; injector temperature 250°C, oven temperature program 100 °C for 1 min, then ramped at 10 °C/min to 280 °C and held 10 min; transfer line temperature 300 °C. MS conditions were as follows: for electron ionization (EI), collection mode profile, 1 microscan, 25 µsec max ion time, range 35–650 µ, source temperature 250 °C, solvent delay 3.00 min, source voltage 70 eV; for chemical ionization (CI), reagent gas NH3 (1.8 mL/min). Samples for CI were run in ddMS2 mode (3 precursor ions) with 1 microscan, 50ms max ion time, 0.450 µ precursor width and dynamic exclusion duration 0.2 min.

EI spectra were manually compared with published data (*Daly et al., 2005*; *Daly et al., 1999*; *Daly et al., 1978*) to identify class and likely ID. A set of known standards was run to give accurate retention times across the range of alkaloids and normalized to literature data using linear regression. Sample retention times were then normalized, and molecular weights were obtained from CI MS1 spectra. These were then directly compared to archival Daly GC-MS data where possible. CI MS2 spectra were also used where possible to confirm functional groups such as alcohols by loss of water, etc. Kovats retention indexes (semi-standard nonpolar) are also provided based on retention times and published indexes for background silicone impurities. Accuracy of index assignments was confirmed based on fatty acid methyl esters from skin lipids present in extracts. Epibatidine coelutes with the lipid methyl palmitoleate and the latter caused a number of false positives in the GC-MS data. We thus reviewed LC-HRMS data at the known elution time relative to a known standard. Epibatidine was only found in one sample in trace quantities and is marked as such.

Samples from Panama and Texas were extracted on separate occasions, then filtered and run in tandem with UHPLC-HESI-MS/MS, following an untargeted metabolomics protocol, with conditions optimized specifically for retention and subsequent identification of alkaloids (*Sedio et al., 2021*). Briefly, for extraction, methanol was evaporated and skins were homogenized with stainless steel beads in a TissueLyser II (QIAGEN Sciences, Germantown, MD, USA) and resuspended in 1800 µL of extraction solvent (9:1 MeOH:H$_2$O). Samples were then extracted for 3 hr at 4 °C in a Thermo-Mixer (Eppendorf US, Enfield, CT, USA), followed by evaporation of the methanol component with a SpeedVac concentrator (Thermo Fisher Scientific, Waltham, MA, USA). Next, samples were freeze-dried with a lyophilizer overnight and resuspended in 500 µL extraction solvent. Resuspended extracts were then filtered and diluted 1:7 in 100% MeOH. The metabolomic extracts were run on a Thermo Fisher Scientific (Waltham, MA, United States) Vanquish Horizon Duo UHPLC system with an Accucore C18 column with 150 mm length, 2.1 mm internal diameter, and 2.6 µm particle size, and a Thermo Fisher Scientific Q Exactive hybrid quadrupole-orbitrap mass spectrometer. The instrumental methods

(e.g. the separation of metabolites by UHPLC, the volumes of buffers and their use in solvent gradients, and the use of heated electrospray ionization [HESI] run in positive ion mode with full-scan MS1 and data-dependent acquisition of MS2 [dd-MS2]) were identical to those described by *Sedio et al., 2021*. A positive reference of 1 μg/μL≥98% (±)-epibatidine dihydrochloride hydrate (Sigma-Aldrich, St. Louis, MO, USA) was included in the run, but injected last in the instrument so as to avoid possible carryover in the column.

Following UHPLC-HESI-MS/MS, chromatographic data were processed using MZmine 3 (v3.9.0; *Schmid et al., 2023*), applying a stringent MS1 noise threshold parameter >100,000 used by other workers (e.g. *Sedio et al., 2021*). So as to avoid additions of false positive metabolite observations, we did not use a gap filling algorithm, a step often used in analysis of chemically homogeneous datasets to backfill overlooked metabolite occurrences. MZmine 3 assigns chromatographic features to putative compounds based on mass-to-charge (*m/z*) ratio and retention time. MZmine 3 feature tables and MS2 data were then uploaded to the Global Natural Products Social Molecular Networking (GNPS) platform (*Wang et al., 2016*) for Feature-Based Molecular Networking (*Nothias et al., 2020*). We used SIRIUS v5.8.6 (*Dührkop et al., 2019*) and CSI:FingerID (*Dührkop et al., 2015*) to infer molecular formulae and predict structures including the elements H, C, N, O, P, and S. CANOPUS was used to classify metabolites (*Dührkop et al., 2021*), following the ClassyFire (*Djoumbou Feunang et al., 2016*) and NPClassifier molecular taxonomies (*Kim et al., 2021*). Only compounds assigned to the alkaloid pathway with an NPClassifier pathway probability score >99% were retained in the feature table, which was generated in R v4.2.2 (*R Development Core Team, 2023*) At >99% confidence, epibatidine was detected in three *S. flotator* skin samples. Its presence was confirmed by manual inspection; the retention time, peak shape, isotope pattern and MS2 are consistent with the epibatidine standard. We note that epibatidine was only abundant enough in one of the three samples to render MS2 fragments.

With respect to the compounds exclusive to the positive reference sample (i.e. not present in the frog skins), at >99% confidence, the algorithms implemented in SIRIUS also predicted annotations consistent with an epibatidine alkaloid for a feature only detected in the positive reference sample, at the levels of most specific class ('epibatidine analogues': ClassyFire) and class and superclass ('pyridine alkaloids'' and 'nicotinic acid alkaloids': NPClassifier). The *m/z* ratio and structural prediction for this feature are consistent with the epibatidine homolog 'homoepibatidine' (*Supplementary file 6*). However, this annotation seems at odds with the true identity of the feature (the retention time is at 0.5 min, the approximate void volume with the highly polar compounds, and the isotope pattern is not correct for Cl, matching better with silicon). Instead, the feature may represent a silicone derivative that, based on results from multiple runs of the instrument (unpublished), we suspect could be an impurity consistently co-occurring with and mistaken for homoepibatidine. In another run, we recovered a feature exclusive to the positive reference sample with annotations identical at all levels to those for our 'homoepibatidine' feature, but with epibatidine's expected *m/z* ratio (~209) and structure (SMILES). In the run we publish here, what is likely this same feature (with an *m/z* ratio of ~209 and annotated as (+/-)-epibatidine by GNPS) was also recovered at the 99% confidence level. Assuming this feature is our positive reference — (+/-)-epibatidine — the molecule was annotated as expected at class and superclass levels ('pyridine alkaloids'' and 'nicotinic acid alkaloids', respectively) but annotated incorrectly at the level of most specific class (as a 'pyrimidinethione'). Our results suggest that SIRIUS sometimes correctly annotates at all pathway levels our (+/-)-epibatidine positive reference.

## Diet identification

Stomach contents were inspected under a stereomicroscope and all prey items identified to order (or family, in the case of Formicidae). Given the low sample sizes in many individuals, we did not conduct statistical comparisons of diet composition across species.

## Analyses

We summarized and plotted data from Ecuadorian and Colombian samples in R v4.3.1 (*R Development Core Team, 2023*) using the packages *ggplot2* (*Wickham, 2016*), *cowplot* v1.1.1 (*Wilke,*

*2020*), and *dplyr* v1.1.2 (*Wickham et al., 2023*). The UHPLC-HESI-MSMS pipeline used on the samples from Panama and Texas allows for higher sensitivity to detect a broader array of compounds compared to our GC-MS methods but has lower retention-time resolution and produces less reliable structural predictions. Furthermore, due to the lack of liquid-chromatography-derived references for poison-frog alkaloids, precise alkaloid annotations from the UHPLC-HESI-MSMS dataset could not be obtained. Therefore, the UHPLC-HESI-MSMS and GC-MS datasets are not directly comparable, and UHPLC-HESI-MSMS data are not included in *Figure 2*. Phylogenies were subsetted from *Wan et al., 2023* using *ape* v5.7.1 (*Paradis and Schliep, 2019*) and *phytools* v1.9.16 (*Revell, 2012*). Co-eluting compounds in the GC-MS and having the same base peak could not be discerned with the parameters we used in the Xcalibur processing method, so we averaged their quantities across the co-eluting compounds. Corrections for mass were not included; we instead opted to provide data from full skins.

## Use of artificial intelligence (AI) and AI-assisted technologies

No AI or AI-assisted technologies were used in the preparation of this manuscript.

## Acknowledgements

We thank Fray Arriaga, Josué Collins (STRI, Panama), Cristian Florez-Pai (FELCA, Colombia), Valentina Gómez-Bahamón, Camilo Isaza (Cafam, Colombia), Roberto Márquez, Daniel Nastacuaz, Pablo Palacios-Rodríguez, Andrea Paz, Santiago Vega, and many others for their assistance in the field. We thank the communities of Laguna de Cube (Esmeraldas) and Laguna de San Pedro (Orellana) in Ecuador for their support and efforts towards conserving local ecosystems. We also thank Kameron T Bell, Nicholas R Andreasen, and Megan M Reid for their help acquiring and processing GC-MS data. We thank Mabel Gonzalez for productive discussions on terminology. UHPLC-HESI-MSMS services were provided by the UT Austin Center for Biomedical Research Support Biological Mass Spectrometry Facility (RRID:SCR_021728), and light microscopy was performed at the Center for Biomedical Research Support Microscopy and Flow Cytometry Facility at UT Austin (RRID:SCR_021756). We acknowledge the Texas Advanced Computing Center (TACC) at The University of Texas at Austin for providing computational resources that contributed to the research results reported within this paper (URL: http://www.tacc.utexas.edu). JLC would like to thank Raineldo Urriola, Isis Ochoa, Lil Marie Camacho, Zurenayka Alain, Roberto Ibáñez, Roberto Cambra, Roberto Borrell, Rivieth De Liones, Félix Rodriguez, and Orelis Arosemena for their warmth and logistical support during his time at STRI. JLC also thanks members of the Anslyn Lab in the UT Austin Chemistry Department for thoughtful discussions about organic chemistry and for generously taking the time to teach him and undergraduate interns as well as share resources. We thank the two reviewers whose feedback greatly improved the manuscript. RDT was supported by an NIH MIRA (R35GM150574), start-up funding from University of California Berkeley, and grants from the Society of Systematic Biologists, North Carolina Herpetological Society, Society for the Study of Reptiles and Amphibians, Chicago Herpetological Society, Texas Herpetological Society, the EEB Program at University of Texas at Austin, National Science Foundation Graduate Research Fellowship Program Graduate Research Opportunities Worldwide (in partnership with USAID), and a National Geographic Young Explorer Grant (#9468–14). Additional support to DCC, RWF, JLC, and RDT was provided by NSF DBI-1556967. JLC received additional support from a Stengl-Wyer Graduate Fellowship from the University of Texas at Austin, and BES received support from Stengl-Wyer Endowment Grant SWG-22–01. RWF and students KSG, JMS, and JRS were supported by NSF DUE-0942345, NSF CHE-1531972, and NSF IOS-1556982.

## Additional information

### Funding

| Funder | Grant reference number | Author |
|---|---|---|
| National Institute of General Medical Sciences | R35GM150574 | Rebecca D Tarvin |

| Funder | Grant reference number | Author |
|---|---|---|
| Society of Systematic Biologists | | Rebecca D Tarvin |
| North Carolina Herpetological Society | | Rebecca D Tarvin |
| Society for the Study of Reptiles and Amphibians | | Rebecca D Tarvin |
| Chicago Herpetological Society | | Rebecca D Tarvin |
| Texas Herpetological Society | | Rebecca D Tarvin |
| University of Texas at Austin | | Rebecca D Tarvin Jeffrey L Coleman |
| University of California Berkeley | | Rebecca D Tarvin |
| National Science Foundation Graduate Research Fellowship | | Rebecca D Tarvin |
| National Geographic Society | #9468-14 | Rebecca D Tarvin |
| National Science Foundation | DBI-1556967 | Rebecca D Tarvin Jeffrey L Coleman David C Cannatella Richard W Fitch |
| The University of Texas at Austin | SWG-22-01 | Brian E Sedio |
| National Science Foundation | DUE-0942345 | Kimberly S Gleason J Ryan Sanders Jacqueline M Smith Richard W Fitch |
| National Science Foundation | CHE-1531972 | Kimberly S Gleason J Ryan Sanders Jacqueline M Smith Richard W Fitch |
| National Science Foundation | IOS-1556982 | Kimberly S Gleason J Ryan Sanders Jacqueline M Smith Richard Fitch |

The funders had no role in study design, data collection and interpretation, or the decision to submit the work for publication.

## Author contributions

Rebecca D Tarvin, Conceptualization, Resources, Data curation, Formal analysis, Supervision, Funding acquisition, Investigation, Visualization, Methodology, Writing – original draft, Project administration, Writing – review and editing; Jeffrey L Coleman, Resources, Data curation, Formal analysis, Funding acquisition, Investigation, Methodology, Writing – original draft, Writing – review and editing; David A Donoso, Resources, Data curation, Investigation, Writing – review and editing; Mileidy Betancourth-Cundar, Karem López-Hervas, Santiago R Ron, Juan C Santos, Resources, Investigation, Writing – review and editing; Kimberly S Gleason, J Ryan Sanders, Jacqueline M Smith, Data curation, Investigation, Writing – review and editing; Brian E Sedio, Resources, Formal analysis, Funding acquisition, Investigation, Writing – review and editing; David C Cannatella, Resources, Supervision, Funding acquisition, Investigation, Writing – original draft, Project administration, Writing – review and editing; Richard W Fitch, Resources, Data curation, Formal analysis, Supervision, Funding acquisition, Investigation, Visualization, Methodology, Writing – original draft, Project administration, Writing – review and editing

## Author ORCIDs
Rebecca D Tarvin (iD) https://orcid.org/0000-0001-5387-7250
Jeffrey L Coleman (iD) https://orcid.org/0000-0002-8156-5948
David A Donoso (iD) https://orcid.org/0000-0002-3408-1457
Mileidy Betancourth-Cundar (iD) http://orcid.org/0000-0003-2368-6028
Karem López-Hervas (iD) http://orcid.org/0000-0002-8296-377X
Brian E Sedio (iD) http://orcid.org/0000-0002-1723-9822
Richard W Fitch (iD) http://orcid.org/0000-0002-4927-725X

## Ethics

Collection in Colombia and Ecuador was performed under permits (COL: Res. 1177 at Universidad de los Andes) and Contrato Marco Acceso a los Recursos Genéticos Nro. 005-14 IC-FAU-DNB/MA (Ecuador). Collection in and export from Panama were performed under Ministerio de Ambiente Permiso de Colecta Científica (No. ARBG-0038-2022) and Permiso de Transferencia de Material Genético y/o Biológico No. PA-01-ARG-096-2022 (Panama), and collection in Texas was performed under scientific research permit SPR-0922-131 issued by the Texas Parks and Wildlife Department. The animal use protocols were approved by the University of Texas at Austin (IACUC AUP-2012-00032, AUP-2021-00042, and AUP-2024-00003) and the Smithsonian Tropical Research Institute (SI-22017). Voucher specimens are deposited in the Museo de Zoología (QCAZ) de Pontificia Universidad Católica del Ecuador (PUCE), the Museo de Historia Natural C.J. Marinkelle (ANDES) at the Universidad de los Andes in Bogotá, Colombia, the Museo de Vertebrados de la Universidad de Panamá (MVUP), and within the Herpetology division of the of the University of Texas at Austin Biodiversity Collections.

Reviewer #1 (Public review): https://doi.org/10.7554/eLife.100011.3.sa1
Reviewer #2 (Public review): https://doi.org/10.7554/eLife.100011.3.sa2
Author response https://doi.org/10.7554/eLife.100011.3.sa3

---

# Additional files

## Supplementary files
Supplementary file 1. Stomach content data for every individual.

Supplementary file 2. A summary of data available on alkaloid detection in undefended lineages of poison frogs.

Supplementary file 3. Collection localities, specimen numbers, size, sex, and summary of alkaloid quantities and diversity for each individual.

Supplementary file 4. Alkaloid-level data for every individual analyzed by GC-MS.

Supplementary file 5. S5a A feature table including information on *Silverstoneia flotator* and *Eleutherodactylus cystignathoides* skin alkaloids; S5b identifying information for samples corresponding to run numbers listed in Table S5a columns.

Supplementary file 6. List of the subset of classes and most specific classes of compounds in *Silverstoneia flotator* annotated as alkaloids ("Alkaloid Pathway" of NPClassifier) at >99% probability, their presence/absence in *Eleutherodactylus cystignathoides*, whether the compound is from one of the classes of lipophilic alkaloids listed in the Daly database, and whether the molecular formula for the metabolite is found in the Daly database.

MDAR checklist

## Data availability
The datasets supporting this article have been uploaded as part of the supplementary material. GC-MS and UHPLC-HESI-MS/MS data are deposited on Global Natural Product Social Molecular Networking as MassIVE datasets under accession numbers MSV000095866 and MSV000094961, respectively. Other raw data are available here in *Supplementary files 1–6*.

The following datasets were generated:

| Author(s) | Year | Dataset title | Dataset URL | Database and Identifier |
|---|---|---|---|---|
| Tarvin RD, Coleman JL, Donoso DA, Betancourth-Cundar M, López-Hervas K, Gleason KS, Sanders JR, Smith JM, Ron SR, Santos JC, Sedio BE, Cannatella DC, Fitch R | 2024 | GNPS-Passive accumulation of alkaloids in non-toxic frogs challenges paradigms of the origins of acquired chemical defenses | https://doi.org/10.25345/C5639KH4H | MassIVE Repository, 10.25345/C5639KH4H |
| Tarvin RD, Coleman JL, Donoso DA, Betancourth-Cundar M, López-Hervas K, Gleason KS, Sanders JR, Smith JM, Ron SR, Santos JC, Sedio BE, Cannatella DC, Fitch R | 2024 | GNPS TarvinETAL2024 Data and Result Files for Silverstoneia flotator and Eleutherodactylus cystignathoides | https://doi.org/10.25345/C57P8TR0D | MassIVE Repository, 10.25345/C57P8TR0D |

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
