## [Editor Report · eLife Assessment]

This study is **important**, with the potential to greatly impact future research on the evolution of chemical defense mechanisms in animals. The authors present **compelling** evidence for the presence of low quantities of alkaloids in amphibians previously thought to lack these toxins. They then integrate these findings with existing literature to propose a four-phase scenario for the evolution of chemical defense in alkaloid-containing poison frogs, emphasizing the role of passive accumulation mechanisms.

---

## [Referee Report · Reviewer #1 (Public review)]

This is a very relevant study, with the potential of having high impact on future research on the evolution of chemical defense mechanisms in animals. The authors present a substantial number of new and surprising experimental results, i.e., the presence in low quantities of alkaloids in amphibians previously deemed to lack these toxins. These data are then combined with literature data to weave the importance of passive accumulation mechanisms into a 4-phases scenario of the evolution of chemical defense in alkaloid-containing poison frogs.

In general, the new data presented in the manuscript are of high quality and high scientific interest, the suggested scenario compelling, and the discussion thorough. Also, the revised version of the manuscript has been carefully prepared with a high quality of illustrations. UI did not detect typos in the text

Understanding that the majority of dendrobatid frogs, including species considered undefended, can contain low quantities of alkaloids in their skin provides an entirely new perspective to our understanding of how the amazing specializations of poison frogs evolved. Although only few non-dendrobatids were included in the alkaloid screening, some of these also included minor quantities of alkaloids, and the capacity of passive alkaloid accumulation may therefore characterize numerous other frog clades, or even amphibians in general.

The overall quality of the work is exceptional. The authors also have done a fantastic job restructuring the manuscript in response to my initial comments, and it is now very clear which new hypotheses are presented and which testable predictions for future studies derive from these hypotheses. This study will be highly influential in informing and guiding future research on toxicity, alkaloid sequestration and resistance, and evolution of aposematism.

---

## [Referee Report · Reviewer #2 (Public review)]

Summary:

This was a well-executed and well-written paper. The authors have provided important new datasets that expand on previous investigations substantially. The discovery that changes in diet are not so closely correlated with the presence of alkaloids (based on the expanded sampling of non-defended species) is important, in my opinion.

Strengths:

Provision of several new expanded datasets using cutting edge technology and sampling a wide range of species that had not been sampled previously. A conceptually important paper that provides evidence for the importance of intermediate stages in the evolution of chemical defense and aposematism.

Weaknesses:

There were some aspects of the paper that I thought could be revised. One thing I was struck by is lack of discussion of the potentially negative effects of toxin accumulation, and how this might play out in terms of different levels of toxicity in different species. Further, are there aspects of ecology or evolutionary history that might make some species less vulnerable to the accumulation of toxins than others? This could be another factor that strongly influences the ultimate trajectory of a species in terms of being well-defended. I think the authors did a good job in terms of describing mechanistic factors that could affect toxicity (e.g. potential molecular mechanisms), but did not make much of an attempt to describe potential ecological factors that could impact trajectories of the evolution of toxicity. This may have been done on purpose (to avoid being too speculative), but I think it would be worth some consideration.

In the discussion, the authors make the claim that poison frogs don't (seem to) suffer from eating alkaloids. I don't think this claim has been properly tested (the cited references don't adequately address it). To do so would require an experimental approach, ideally obtained data on both lifespan and lifetime reproductive success.

Update: Revised version: The authors carefully addressed the comments and suggestions on the first draft of the manuscript. In my opinion, these revisions were sufficient and the authors have adequately addressed the previously noted weaknesses in the manuscript.

---

## [Author Response]

The following is the authors’ response to the original reviews.

**Public Reviews:**

**Reviewing editor:**
The biological significance of the results presented in this manuscript is the potential absence of active sequestration mechanisms in certain species, leading to variation in their ability to transport and store specific compounds, such as alkaloids. The concept of passive accumulation is introduced as an evolutionary intermediate between toxin consumption and sequestration.I agree with the reviewers' comments on the limitations of the current manuscript. Additionally, I'd like to raise a point about combining data from LC/MS and GC/MS as these techniques have different sensitivities. GC-MS excels in annotation, allowing for confident identification of detected compounds. However, it may have limitations in the number of extractable substances. Conversely, LC-MS/MS offers a broader range of detectable substances, but annotation can be more challenging. While methods to bridge this gap exist, the current approach might not fully account for the potential influence of the analysis equipment on the observed differences in alkaloid numbers between the Texas and Panama samples analyzed by LC-MS/MS. To address this, consider including data from both methods (if possible) to gain a more comprehensive understanding of the alkaloid profiles. Alternatively, analyzing the Texas and Panama samples with GC-MS could be considered for a more focused comparison with the other samples.

Thank you for the suggestion. Unfortunately, we do not have GC-MS data for the Texas and Panama samples. While the strength of these two datasets is that they present two independent lines of data corroborating that “undefended” frogs have detectable alkaloid levels, we have more explicitly made clear for readers that the datasets should not be compared directly. We reviewed the text to check that we carefully acknowledge in the manuscript the higher sensitivity of our LC-MS assay, and we added more detail about the differences between the two assay types (section 4d): “The UHPLC-HESI-MSMS pipeline used on the samples from Panama and Texas allows for higher sensitivity to detect a broader array of compounds compared to our GC-MS methods, but has lower retention-time resolution and produces less reliable structural predictions. Furthermore, due to the lack of liquid-chromatography-derived references for poison-frog alkaloids, precise alkaloid annotations from the UHPLC-HESI-MSMS dataset could not be obtained. Therefore, the UHPLC-HESI-MSMS and GC-MS datasets are not directly comparable, and UHPLC-HESI-MSMS data are not included in Fig. 2”. We have also revised the asterisk accompanying the table to further reinforce that alkaloid numbers between the two assay types should not be compared. It now states: “Note that the UHPLC-HESI-MS/MS and GC-MS assays differed in both instrument and analytical pipeline, so “Alkaloid Number” values from the two assay types should not be compared to each other directly”. We further point out differences between the two assay types in section 2b: “Similarly, the analysis of UHPLC-HESI-MS/MS data was untargeted, and thus enables a broader survey of chemistry compared to that from prior GC-MS studies.”

Finally, we point out that the output from the analytical pipeline for UHPLC-HESI-MSMS annotates compounds as “alkaloids,” using broader criteria than the targeted GC-MS component of our study. In an effort to make the datasets more comparable, at least conceptually, we now include an assessment of which alkaloids identified by UHPLC-HESI-MSMS match known molecular formulae and structural classes in frogs see Table S6 and revised text on lines 335-343 and 410-415.

**Reviewer #1 (Public Review):**
This is a very relevant study, clearly with the potential of having a high impact on future research on the evolution of chemical defense mechanisms in animals. The authors present a substantial number of new and surprising experimental results, i.e., the presence in low quantities of alkaloids in amphibians previously deemed to lack these toxins. These data are then combined with literature data to weave the importance of passive accumulation mechanisms into a 4-phases scenario of the evolution of chemical defense in alkaloid-containing poison frogs.In general, the new data presented in the manuscript are of high quality and high scientific interest, the suggested scenario compelling, and the discussion thorough. Also, the manuscript has been carefully prepared with a high quality of illustrations and very few typos in the text. Understanding that the majority of dendrobatid frogs, including species considered undefended, can contain low quantities of alkaloids in their skin provides an entirely new perspective to our understanding of how the amazing specializations of poison frogs evolved. Although only a few non-dendrobatids were included in the GCMS alkaloid screening, some of these also included minor quantities of alkaloids, and the capacity of passive alkaloid accumulation may therefore characterize numerous other frog clades, or even amphibians in general.

Thank you for the kind evaluation.

While the overall quality of the work is exceptional, major changes in the structure of the submitted manuscript are necessary to make it easier for readers to disentangle scope, hypotheses, evidence and newly developed theories.

Based on reviewer comments, we revised the manuscript structure substantially to make the different aspects of the paper more readily identifiable to readers. Specifically we moved the content of Figure 2 into a new section in the introduction. We also added more introductory text to better introduce the main ideas of the new model and to summarize the scope and aim of the paper. We reorganized the result section headings and moved Figure 1 (now Fig. 3) down into section 2c.

**Reviewer #2 (Public Review):**
Summary:This was a well-executed and well-written paper. The authors have provided important new datasets that expand on previous investigations substantially. The discovery that changes in diet are not so closely correlated with the presence of alkaloids (based on the expanded sampling of non-defended species) is important, in my opinion.Strengths:Provision of several new expanded datasets using cutting edge technology and sampling a wide range of species that had not been sampled previously. A conceptually important paper that provides evidence for the importance of intermediate stages in the evolution of chemical defense and aposematism.

Thank you for kind comments.

Weaknesses:There were some aspects of the paper that I thought could be revised. One thing I was struck by is the lack of discussion of the potentially negative effects of toxin accumulation, and how this might play out in terms of different levels of toxicity in different species.

Thank you for the suggestion. We now explicitly address the possible negative effects of toxin accumulation and how costs may play out with respect to varying levels of chemical defense among different organisms, including poison frogs. We note early on that, “short-term alkaloid feeding experiments (e.g., Daly et al., 1994; Sanchez et al., 2019) demonstrate that both defended and undefended dendrobatids can survive the immediate effects of alkaloid intake, although the degree of resistance and the alkaloids that different species can resist vary'' (section 2c), and we address the sparse literature suggesting some species-level variation in alkaloid resistance in frogs. Later, we make the point that, “origins of chemical defenses are also shaped by the cost of resisting and accumulating toxins, which can change over evolutionary time as animals adapt to novel relationships with toxins” (section 2d). We broadly discuss costs of target-site resistance, a common mode of molecular resistance in poison frogs and other animals, and compensatory molecular adaptations that offset the costs. We also discuss examples from the literature of negative effects of high levels of resistance and toxin accumulation that are not completely offset. We also note that to the best of our knowledge, potential lifetime fitness costs to alkaloid consumption by dendrobatids have not been evaluated.

Further, are there aspects of ecology or evolutionary history that might make some species less vulnerable to the accumulation of toxins than others? This could be another factor that strongly influences the ultimate trajectory of a species in terms of being well-defended. I think the authors did a good job in terms of describing mechanistic factors that could affect toxicity (e.g. potential molecular mechanisms) but did not make much of an attempt to describe potential ecological factors that could impact trajectories of the evolution of toxicity. This may have been done on purpose (to avoid being too speculative), but I think it would be worth some consideration.

We agree that other factors can influence the trajectory of chemical defense. We incorporated these ideas into the new section 2d, which provides a somewhat brief overview of ecological factors that could influence the origins of chemical defense, the physiological costs of toxin resistance and accumulation, and some of the possible eco-evo factors that shape chemical defense once it evolves.

In the discussion, the authors make the claim that poison frogs don't (seem to) suffer from eating alkaloids. I don't think this claim has been properly tested (the cited references don't adequately address it). To do so would require an experimental approach, ideally obtained data on both lifespan and lifetime reproductive success.

We agree with the reviewer that more data are necessary to make this broad claim, which we have removed. We revised this to state: “regardless, it is clear that all or nearly all dendrobatid poison frogs consume alkaloid-containing arthropods as part of their regular diet” (section 2c). We then expand on this statement with data from short-term experimental work that support the notion that at least some dendrobatids are resistant (i.e., can survive) the immediate effects of alkaloids. We also point out later in the manuscript that, “as far as we are aware, the possible lifetime fitness costs (e.g., in reproductive success) of alkaloid consumption in dendrobatids have not been measured” (section 2d).

**Recommendations for the authors:**

**Reviewer #1 (Recommendations For The Authors):**
While in general I am very open to "unorthodox" ways to write a manuscript (i.e., differing from the standard structure intro-methods-results-discussion) I feel there is much room for improvement in this case. When reading the manuscript line by line, I was several times totally uncertain about the scope and content of the original data in the manuscript. It is too often unclear which of the outlined theories are new and why they are presented, which hypotheses were tested and why, which data were newly obtained, which technological improvements led to the novel and surprising results, and why no alternative hypotheses are tested. I feel the authors need to fundamentally reconsider the structure of the manuscript - which does not mean everything needs to be rewritten, but some major reshuffling of paragraphs from one section to the other may already lead to substantial improvement. I will in the following list (not ordered by priority) different issues that I encountered, without always providing a specific suggestion for improvement - please come up with an improved structure that removes these issues in one way or the other!

Thank you for the suggestions. We did our best to improve the structure of the paper. Specifically, we substantially revised the introduction to provide a clearer background of the ideas leading up to the new evolutionary model. We moved most of what was previously figure 2 (now Fig. 1) into an earlier part of the introduction in the main text. We moved what was previously figure 1 (now Fig. 3) to much later in the discussion (section 2c). We attempted to clarify and separate throughout the text the new data from existing data. Please see our responses below for additional details.

Line 42-45: Please provide a reference on this statement on traversing adaptive landscapes.

We added the following reference: Martin, CH and PC Wainwright. 2013. Multiple fitness peaks on the adaptive landscape drive adaptive radiation in the wild. *Science* 339: 208-211. https://doi.org/10.1126/science.1227710

Line 50: Why are these phases "likely" to occur? - no evidence is presented for this hypothesized high likelihood. Presenting this scenario already in the second paragraph of the intro is very weird. Are these really the only possible phases? Wouldn't it be possible to come up with totally different scenarios? In my opinion, this specific four-phase scenario should be more clearly labelled as a novel theory presented in this paper, and perhaps it should come much later in the introduction.

Thank you for the suggestion. We moved this paragraph down into a new subsection of the introduction. We also revised the language to clarify that the model is a new evolutionary theory based on new and existing ideas.

Line 51: Here you use for the first time the term "elimination". While it is intuitively clear what is meant by it, there still could be different meanings. The alkaloids could simply be passively excreted, or they could be actively biochemically decomposed. Later in the Discussion the authors imply that elimination requires some kind of metabolic process, but this perhaps should be made clearer already in the introduction.

We now spend more time in the introduction describing pharmacokinetics as well as the terms we used (including elimination), which are slightly modified from terms in pharmacokinetics.

Figure 1. I have major concerns about this figure. I found the figure very confusing, and the authors really need to reconsider and modify (simplify) it. The figure caption starts with "Major processes involved..." as if this was established textbook knowledge rather than a totally hypothetical illustration of how different factors (sequestration, elimination....) can lead to defended or undefended phenotypes. Only later on in the caption it becomes clear this is just a suggestion/hypothesis/model: "we hypothesize...".

We revised the figure (now Fig. 3) and its legend. It now starts with the following text: “Hypothesized physiological processes that interact to determine the defense phenotype.” We also simplify the figure by removing two lines and recoding the table (see comment below).

Secondly, the way the graph is drawn suggests some kind of experimental result where specific evolutionary pathways lead to very specific degrees of "defendedness", recognizable by the points on the right axis stacked very precisely one above the other. Do you really want to imply that you want to suggest such a specific model, where particular accumulation/intake/elimination rates lead to exactly these outcomes? Also, wouldn't it be possible to somewhat simplify the categories in the table? Again, why so specific, is there any experimental evidence for it? Why sometimes 1 plus, 2 plus, 3 plus? Wouldn't it be better to just suggest categories such as strong, weak and absent?

We simplified the figure by removing the secondary (dashed) passive accumulation and active sequestration lines. We also changed the + signs to “low,” “med,” or “high” and tried to simplify the text in the figure and in the legend.

Line 101-103: "We propose ..." Here, as the concluding statement of the introduction, the authors suggest a very general hypothesis which seems rather disconnected from the four-phase model and from the experimental results. Here, at the latest, I would have expected to learn (1) what the overall scope of the paper is, (2) which kind of approaches were followed and which novel experimental results will be presented in the following, and (3) how the experimental results will be used to derive a new theory / novel. Again, it is obvious that the scope of the paper is broader than testing just a single and narrow hypothesis, but rather to support and develop a broader theory and evolutionary model, but this should be clear to readers once they arrive at this line.

Thank you for the suggestion. We added a paragraph to the end of the first section of the introduction that outlines the content of the rest of the paper. We also reorganized some of the subheadings to make the flow of ideas and the source of data in each subsection clearer. We split up and moved what was previously in section 2a into parts of the introduction and discussion. We moved the results text about diet and the discussion about resistance to section 2a, to better provide data and discussion of phases 1 and 2.

Figure 2. My opinion on this figure is much less strong than on Fig. 1. However, the authors may want to reconsider whether it really makes sense to here show all the historical trees and theories (which are not really systematically reviewed in the text) or if they maybe wish to go on with panel D only (the most recent tree and scenario which is also used to consistently for further discussion in the manuscript).

We moved the content from Fig. 2A–C to the main text (now section 1b) and narrowed the focus of Fig. 2 (now Fig. 1) to what was previously panel 2D.

Results and Discussion: The whole section on phases 1 to 2 is not based on any new results. This is OK (as I said, I have no problems with "unorthodox" manuscript structure) but it should be clearer to readers why this is presented here and what it represents. A new theory? A recapitulation of textbook knowledge? Something necessary to later understand the experimental results?

We split up and moved what was previously in section 2a into parts of the introduction and discussion. Now, section 2a still focuses on phases 1 and 2 but presents the diet data from our study (phase 1) and a review of known resistance mechanisms (phase 2; previously in the discussion section).

Line 168. Here we have arrived at the "core" of the paper, that is, the actual experimental results. Surprisingly, you find alkaloids in dendrobatids usually considered "undefended". This is great, surprising and of high importance. However, I am missing at least some technical/methodological discussion about this finding, except for the statement that it was based on GCMS. Why have previous studies not detected these alkaloids? Did you use particularly sensitive GCMS instruments? Did you look more in depth than it was done in previous studies? Can you totally exclude these contaminations/artefacts?

We added the following paragraph to section 2b: “The large number of structures that we identified is in part due to the way we reviewed GC-MS data: in addition to searching for alkaloids with known fragmentation patterns, we also searched for anything that could qualify as an alkaloid mass spectrometrically but that may not match a previously known structure in a reference database. Similarly, the analysis of UHPLC-HESI-MS/MS data was untargeted, and thus enables a broader survey of chemistry compared to that from prior GC-MS studies. Structural annotations in our UHPLC-HESI-MS/MS analysis were made using CANOPUS, a deep neural network that is able to classify unknown metabolites based on MS/MS fragmentation patterns, with 99.7% accuracy in cross-validation (Dührkop et al., 2021).” We also moved the paragraph on contamination from the methods section into section 2b.

Line 169. This sentence (and several others in the subsequent paragraphs) do a poor job in explaining the taxon and specimen sampling. The particular sentence in this line is unclear: Did you include 27 species of dendrobatids AND IN ADDITION representatives of the main undefended clades, or did these 27 species INCLUDE representatives of the main undefended clades?

We now present a brief overview of sampling in the last paragraph of the introduction (section 1c). We clarified sampling of the species: “In total we surveyed 104 animals representing 32 species of Neotropical frogs including 28 dendrobatid species, two bufonids, one leptodactylid, and one eleutherodactylid (see Methods). Each of the major undefended clades in Dendrobatidae (Fig. 1, Table 1) is represented in our dataset, with a total of 14 undefended dendrobatid species surveyed.” We also reviewed and clarified similar language in other places in the text (e.g., section 2b).

Line 177. "undefended lineages" - of dendrobatids or of frogs in general? Given that you also include non-dendrobatids.

Dendrobatids. The sentence now reads “Overall, we detected alkaloids in skins from 13 of 14 undefended dendrobatid species included in our study, although often with less diversity and relatively lower quantities than in defended lineages (Fig. 2, Table 1, Table S3, Table S4).”

Line 188: "defe" should probably changed to "defended"?

Corrected.

Table 1. The taxon sampling clearly focuses on dendrobatids, with only a few other taxa. This is fine, however, it does not allow to test the hypothesis that something "special" predisposes dendrobatids to passive accumulation and alkaloid resistance. For this, a wider taxon sampling of other frog families would have been necessary to have a larger number of "control" data. Again, this is fine for the purpose of the study and is discussed later (line 399) but only very briefly. I feel it should be mentioned earlier on.

Thank you for the suggestion. We now address this point earlier in the manuscript so that readers will not have the impression that there are sufficient data to infer that dendrobatids are predisposed to passive accumulation. We propose several phylogenetic alternatives, making it clear that determining the number and timing of origins of passive accumulation is not possible with our data (section 2c), ultimately noting that “discriminating a single origin [of passive accumulation] – no matter the timing – from multiple ones would require better phylogenetic resolution and more extensive alkaloid surveys, as we only assessed four non-dendrobatid species”.

**Reviewer #2 (Recommendations For The Authors):**
P2L60 - The description of figure 1 is somewhat confusing, as it first focuses on the graph in the bottom panel, then moves to describing aspects of the table (top panel), then back to the graph. I think it might make more sense to describe these two panels separately and in order.

Thank you for the suggestion. We revised the figure (now Fig. 3) and its legend for clarity.

P3L94 - Saying that three transitions makes this group "ideal" for studying complex phenotypic transitions is a bit hyperbolic, in my opinion. I suggest toning down this description.

Thank you for the suggestion. We changed “ideal” to “suitable.”

P3L101 - "We propose that changes in toxin metabolism through selection on mechanisms of toxin resistance likely play a major role in the evolution of acquired chemical defenses." This hypothesis appears to be a combination of earlier ideas, with a somewhat different emphasis. The authors acknowledge this and go through some of the earlier ideas, in the legend of figure 2. I would have preferred to see more discussion of this (particularly with reference to the history of the idea in reference to poison frogs) in the main body of the text.

Thank you for the suggestion. We now more extensively discuss these prior studies in the introduction (section 1b and 1c). We also revised this figure (now Fig. 1) to focus on what was previously figure 2 panel D.

P3L102 - Figure 2 - the phrase "Resistance to consuming some alkaloids" seems inappropriate - perhaps "Resistance to alkaloid poisoning after consumption" (or something similar) would be more accurate?

We changed this to “Low alkaloid resistance”.

P4L153 - "Accumulation of alkaloids in skin glands could help to prevent alkaloids from reaching their targets". This could be true, but why would skin glands be a preferred location of sequestration to avoid toxicity? The authors should explain why such glands would be particularly likely to serve as places of sequestration.

Thank you for pointing out this ambiguity. We decided to remove our discussion of sequestration into skin glands, because it is challenging to discuss this process in toxin resistance without too much speculation.

P4L154 - "Although direct evidence is lacking, some poison frogs may biotransform alkaloids into less toxic forms until they can be eliminated from the body, e.g., using cytochrome p450s". This would seem to contradict the argument of this process being a precursor to accumulating effective toxins.

We agree that these processes seem contradictory. However, a few papers are starting to suggest that metabolic detoxification may be initially useful for lineages that eventually evolve toxin sequestration. This is because detoxification or elimination (clearance) of toxins allows increased intake of toxins. Because there is some delay in the removal of toxins from an animal’s body, increased consumption ultimately leads to higher toxin exposure and possible toxin diffusion into various body cavities, which can increase selective pressure to evolve other kinds of resistance mechanisms. This pattern was shown in an experiment with toxin-resistant fruit flies (Douglas et al., 2022). Many toxin-sequestering species still metabolize some toxins even if they sequester the majority – as we argue, the defense phenotype is the result of a balance among intake, elimination, and accumulation, all of which can interact simultaneously. In poison frogs specifically there is some evidence that p450s are upregulated after toxin consumption (Caty et al. 2019). One possible prediction is that the type of resistance that an animal has changes as toxin sequestration evolves. We talk a bit more about these patterns in section 2e.

P5L186 - Table 1 legend - change "defe" to "defended"

Corrected.

P12L414 - "do not appear to suffer substantially from doing so as it is part of their regular diet". I don't think this claim has been properly tested, as of yet. It would require looking at the effects of a diet with and without toxins over the lifespan of the frogs, and the impact of that difference on both survival and fertility.

Reviewer 1 also made this important observation, which we address above.

P12L432 - "for toxin-resistant organisms, there is little cost to accumulating a toxin, yet there may be benefits in doing so." Yet toxin resistance may itself be a continuous trait, so there may be a cost that depends on the degree of toxin resistance. I don't see why the authors are proposing toxin resistance as a discrete trait when their main point is that toxin accumulation is not.

We agree and removed this statement.